# Type 2 diabetes disrupts circadian orchestration of lipid metabolism and membrane fluidity in human pancreatic islets

Volodymyr Petrenko[1,2,3,4☯], Flore Sinturel[1,2,3,4☯], Ursula Loizides-Mangold[1,2,3,4], Jonathan Paz Montoya[5,6], Simona Chera[1,2,3,4,7], Howard Riezman[8], Charna Dibner [1,2,3,4]*

1 Thoracic and Endocrine Surgery Division, Department of Surgery, University Hospital of Geneva, Geneva, Switzerland, 2 Department of Cell Physiology and Metabolism, Faculty of Medicine, University of Geneva, Geneva, Switzerland, 3 Diabetes Center, Faculty of Medicine, University of Geneva, Geneva, Switzerland, 4 Institute of Genetics and Genomics in Geneva (iGE3), Geneva, Switzerland, 5 Proteomics Core Facility, EPFL, Lausanne, Switzerland, 6 Institute of Bioengineering, School of Life Sciences, EPFL, Lausanne, Switzerland, 7 Department of Clinical Science, University of Bergen, Bergen, Norway, 8 Department of Biochemistry, Faculty of Science, NCCR Chemical Biology, University of Geneva, Geneva, Switzerland

☯ These authors contributed equally to this work.
* charna.dibner@hcuge.ch

**Data Availability Statement:** All relevant data are within the paper and its supporting files.

**Funding:** This work was funded by Swiss National Science Foundation grant 310030_184708/1 (CD);

## Abstract

Recent evidence suggests that circadian clocks ensure temporal orchestration of lipid homeostasis and play a role in pathophysiology of metabolic diseases in humans, including type 2 diabetes (T2D). Nevertheless, circadian regulation of lipid metabolism in human pancreatic islets has not been explored. Employing lipidomic analyses, we conducted temporal profiling in human pancreatic islets derived from 10 nondiabetic (ND) and 6 T2D donors. Among 329 detected lipid species across 8 major lipid classes, 5% exhibited circadian rhythmicity in ND human islets synchronized in vitro. Two-time point-based lipidomic analyses in T2D human islets revealed global and temporal alterations in phospho- and sphingolipids. Key enzymes regulating turnover of sphingolipids were rhythmically expressed in ND islets and exhibited altered levels in ND islets bearing disrupted clocks and in T2D islets. Strikingly, cellular membrane fluidity, measured by a Nile Red derivative NR12S, was reduced in plasma membrane of T2D diabetic human islets, in ND donors' islets with disrupted circadian clockwork, or treated with sphingolipid pathway modulators. Moreover, inhibiting the glycosphingolipid biosynthesis led to strong reduction of insulin secretion triggered by glucose or KCl, whereas inhibiting earlier steps of de novo ceramide synthesis resulted in milder inhibitory effect on insulin secretion by ND islets. Our data suggest that circadian clocks operative in human pancreatic islets are required for temporal orchestration of lipid homeostasis, and that perturbation of temporal regulation of the islet lipid metabolism upon T2D leads to altered insulin secretion and membrane fluidity. These phenotypes were recapitulated in ND islets bearing disrupted clocks.

the Vontobel Foundation (CD); the Novartis Consumer Health Foundation (CD); European Foundation for the Study of Diabetes EFSD/Novo Nordisk A/S Programme for Diabetes Research in Europe 2020 (CD); the Swiss Life Foundation (CD); the Olga Mayenfisch Foundation (CD); Leenaards Foundation (CD); the Swiss Cancer Research foundation (CD); Fondation pour l'innovation sur le cancer et la biologie (CD); Ligue pulmonaire genevoise (LPGE, CD); Bo and Kerstin Hjelt Foundation for diabetes type 2 (U L-M. and VP); Young Independent Investigator Grant SGED/SSED (VP and FS). SC was supported by funds from the Research Council of Norway (NFR 251041) and Novo Nordic Foundation (NNF15OC0015054 and NNF21OC0067325). HR was supported by the Swiss National Science Foundation grant 310030_184949 and the NCCR Chemical Biology (51NF40-185898). The funders had no role in study design, data collection and analysis, decision to publish, or preparation of the manuscript.

**Competing interests:** The authors have declared that no competing interests exist.

**Abbreviations:** CerS2, ceramide synthase 2; DAL, differentially abundant lipid; GlcCer, glucosylceramide; GP, generalized polarization; GSIS, glucose-stimulated insulin secretion; HexCer, hexosylceramide; HexDHCer, hexosyldihydroceramide; HPRT, hypoxanthine-guanine phosphoribosyltransferase; MUFA, monounsaturated fatty acid; ND, nondiabetic; PC, phosphatidylcholine; PE, phosphatidylethanolamine; PI, phosphatidylinositol; PS, phosphatidylserine; PUFA, polyunsaturated fatty acid; RT, room temperature; SFA, saturated fatty acid; siRNA, small interfering RNA; T2D, type 2 diabetes.

## Introduction

Internal circadian clocks have evolved in most living beings to allow anticipation of daily light changes due to the Earth rotation. In mammals, this body timekeeping system relies on a central pacemaker residing in paired suprachiasmatic nuclei in the hypothalamus and multiple peripheral clocks in the organs [1,2]. It comprises myriads of cell-autonomous oscillators operative in most cells that ensure temporal orchestration of all aspects of physiology and metabolism [3,4]. At the same time, a dramatic rise in cardiometabolic diseases including obesity and type 2 diabetes (T2D) worldwide has been associated with the 24/7 lifestyle of our society that leads to chronic desynchrony between internal circadian system and external synchronizing cues (light, eating) dubbed circadian misalignment [3,5,6].

In pancreatic islets, studies in mouse models revealed that functional molecular oscillators are indispensable for absolute and temporal regulation of insulin and glucagon secretion [7–9], and for compensatory β-cell regeneration [10]. Surprisingly, the circadian clocks in neighboring α- and β-cells are not phase aligned, and they exhibit cell-specific circadian response to physiologically relevant synchronizers such as adrenalin, glucagon, GLP1, or somatostatin [8,11]. Consistently, clock-deficient mice show severe perturbations of glucose and lipid metabolism that are exacerbated upon islet-specific clock disruption and lack of β-cell regenerative capacity following massive ablation [7,10,12]. In humans, cell-autonomous clocks operative in α- and β-cells orchestrate the rhythmic pattern of proinsulin, insulin, and glucagon secretion [13–15]. Disruption of functional oscillators in human islet cells from ND (nondiabetic) donors, mediated by small interfering RNA (siRNA) targeting of CLOCK, resulted in strongly diminished absolute levels and perturbed rhythmicity of basal insulin secretion exerted via reduced secretory granule docking and exocytosis [13,15]. Most strikingly, our recent study reveals that the circadian clockwork is compromised in human α- and β-cells in T2D. Clock perturbation in T2D islets was paralleled with altered temporal profiles of insulin and glucagon secretion [15].

Lipid metabolites are involved in energy homeostasis, membrane function, and signaling, thus playing essential roles in regulating body metabolism and in pathophysiology of metabolic disorders [16–20]. Mass spectrometry–based shotgun lipidomics allows quantification of over 1,000 phospholipids, sphingolipids, and triacylglycerides with high accuracy [21,22]. Using this powerful approach, it has been demonstrated that in mouse liver, a large portion of lipid species across all major lipid classes display a circadian rhythm, and this rhythmicity is driven by both circadian clocks and feeding [23]. In humans, metabolomics and lipidomics of serial blood samples suggested diurnal profiles for a wide panel of metabolites and lipids [24–26]. Lipidomics of serial human skeletal muscle biopsies obtained across 24 h [27] revealed that about 20% of the lipid species across all major lipid classes display a circadian rhythm in ND patients [28,29]. Strikingly, the rhythmicity of the lipid metabolites has been preserved in human skeletal myotubes differentiated and synchronized in vitro, highlighting that primary cells synchronized in vitro represent invaluable models for studying temporal regulation of lipid metabolism in humans [27]. Oscillating lipids in both skeletal muscle tissue and in cultured myotubes were not only limited to energy-controlling storage lipids such as triglycerides, but also comprised membrane and signaling lipids of different cellular compartments [29]. In line with these findings, parallel RNA-seq analyses based on the same experimental design suggested that key enzymes regulating lipid biosynthesis and glucose metabolism in the skeletal muscle exhibited rhythmic profiles [30]. Furthermore, our earlier works suggest that cell-autonomous circadian oscillators are indispensable for the proper coordination of glucose uptake and temporal lipid profiles in human muscle, since glucose uptake was reduced and the

lipid oscillations were blunted upon *siClock*-mediated disruption of the skeletal myotube oscil-lator [29,30].

Dysregulation of lipid metabolism plays a key role in pathophysiology of metabolic diseases. Lipidomic approaches have pointed to novel mechanistic insights into pathophysiology of obe-sity and T2D [16,17,20,26,31–34]. The pattern of lipid alterations was tissue and disease spe-cific, allowing to define distinct lipid signatures associated with obesity or T2D [35]. The blood levels of ceramides species and 1-deoxysphingolipids have been proposed as T2D biomarker candidates or therapeutic targets [36–41]. Whereas the roles of lipid metabolites in β-cell func-tion and dysfunction upon T2D development have been raised in several studies conducted in immortalized cell lines [42,43] but also in mouse models and in humans [44], no data on human islet lipidomics and its regulation by the circadian system have been provided so far. To fill this gap, we aimed to uncover molecular determinants of circadian regulation of lipid homeostasis in human pancreatic islets under physiological conditions and upon T2D. Employing lipidomic approaches, we demonstrate the circadian rhythmicity of phospho- and sphingolipids in human pancreatic islets from ND donors synchronized in vitro. Most impor-tantly, we provide a novel link between disruption of circadian clock, temporal coordination of lipid metabolism in human pancreatic islet, and islet dysfunction upon T2D in humans, highlighting both molecular oscillator and sphingolipid metabolites as important therapeutic targets for metabolic diseases.

## Results

### Circadian lipidomics of human pancreatic islets synchronized in vitro

To examine the role of cell-autonomous circadian oscillators operative in human pancreatic islets in temporal orchestration of the islet lipid homeostasis, we conducted lipidomic analysis of intact human pancreatic islets synchronized in vitro. Islets obtained from 6 ND donors (see Table 1 for donor characteristics) were synchronized by a forskolin pulse and collected across 24 h according to the experimental design presented in Fig 1A. Rhythmic expression of key core clock transcripts validated efficient in vitro synchronization of human pancreatic islets (S1 Fig). Out of 711 measured lipids, a total of 410 lipid species clustered in 8 major lipid clas-ses were detected across all donors (S2A and S2B Fig and S1 Data). The percentage of lipids exhibiting diurnal oscillations according to the METACYCLE algorithm varied from 0.98% to 14.88% among the donors (Fig 1B and S2 Data). A peak of accumulation of rhythmic lipids was observed 12 h to 16 h following in vitro synchronization in most of the donors (Figs 1C and S2C). When the lipid species were clustered by lipid class, the distribution of the rhythmic lipids indicated that certain lipid classes were preferentially subject to circadian oscillations, although this distribution varied across the donors (S2D Fig). Phosphatidylinositol (PI) lipids were particularly enriched among the cyclic species in all donors (Figs 1B and S2D), even when the overall number of rhythmic lipid species was low, like in donor 5. We further investi-gated the abundance of different PI lipids throughout the circadian cycle. Lyso-, diacyl-, and ether-containing PIs displayed a common pattern of oscillation with a peak at 12 h and a nadir at 32 h after synchronization and up to 2-fold circadian amplitude (Fig 1D). We identified 3 individual lipids significantly rhythmic ($p < 0.05$) in at least 3 islet batches out of 6, all of them belonging to the PI lipid class: PI(O-)44:4, PI28:3, and PI40:2 (Figs 1E and S2E–S2G), the for-mer being much more abundant than the others in this lipid class (S2H Fig). Noteworthy, the degree of desaturation of these lipids influenced their temporal profiles. Indeed, while PI satu-rated in their fatty acyl chains (referred to as saturated fatty acids (SFAs)) exhibited circadian rhythmicity with a single peak of abundance at 12 h after in vitro synchronization, the MUFA and PUFA (respectively monounsaturated and polyunsaturated) PIs displayed profiles closer

**Table 1. Human donor characteristics.**

| Donor no. | Sex | Age (years) | BMI (kg/m$^2$) | HbA$_{1c}$ | Islet purity (%) | Biopsy source |
|---|---|---|---|---|---|---|
| Characteristics of non-T2D (ND) islet donors | | | | | | |
| ND 1[a] | M | 51 | | <6.0 | 63 | *BCN* |
| ND 2[a] | M | 54 | 28.5 | <6.0 | 85 | *HUG* |
| ND 3[a] | F | 59 | 23.7 | <6.0 | 70 | *HUG* |
| ND 4[a] | F | 51 | 44.1 | <6.0 | 60 | *HUG* |
| ND 5[a,c] | F | 47 | 33.4 | <6.0 | 60 | *HUG* |
| ND 6[a] | M | 49 | 26.2 | <6.0 | 95 | *HUG* |
| ND 7[b] | M | 20 | 23.5 | <6.0 | 84 | *HUG* |
| ND 8[b] | M | 47 | 23.4 | <6.0 | 80 | *HUG* |
| ND 9[b,d] | F | 59 | 25.9 | <6.0 | 88 | *HUG* |
| ND 10[b] | F | 76 | 19.3 | <6.0 | 80 | *UAL* |
| ND 11[c,e] | M | 23 | 24 | 5 | 90 | *UAL* |
| ND 12[c,e] | F | 46 | 19 | 5.8 | 85–90 | *Prodo* |
| ND 13[c,e] | M | 47 | 31 | 5.1 | 85 | *UAL* |
| ND 14[c] | M | 45 | 29.7 | 5.5 | 90 | *UAL* |
| ND 15[c] | M | 46 | 27.2 | <6.0 | 75 | *HUG* |
| ND 16[c,e] | F | 53 | 22.7 | 4.8 | 90 | *Prodo* |
| ND 17[c] | M | 59 | 25.6 | <6.0 | 75 | *HUG* |
| ND 18[c] | F | 55 | 21.9 | <6.0 | 60 | *HUG* |
| ND 19[d] | M | 49 | 18.6 | <6.0 | 62 | *HUG* |
| ND 20[d] | M | 60 | 24.2 | <6.0 | 83 | *HUG* |
| ND 21[d] | M | 29 | 23 | 5.1 | 85 | *Prodo* |
| ND 22[d] | F | 24 | 25.2 | <6.0 | 90–95 | *Prodo* |
| ND 23[e] | M | 48 | 19 | 5.3 | 85–90 | *Prodo* |
| ND 24[e] | F | 35 | 26.7 | 3.8 | 95 | *UAL* |
| ND 25[e] | M | 31 | 34.9 | 5.7 | 80 | *UAL* |
| **M = 15, F = 10** | | **46.52 ± 13.21** | **25.87 ± 5.78** | **<6.0** | | |
| Characteristics of T2D islet donors | | | | | | |
| T2D 1[b] | M | 51 | 35.6 | 7.1 | 90 | *Prodo* |
| T2D 2[b] | M | 59 | 27.7 | 6.5 | 85 | *Prodo* |
| T2D 3[b,c] | M | 51 | 35.3 | 8.6 | *NA* | *UAL* |
| T2D 4[b,c] | M | 65 | 21.8 | 9.9 | *NA* | *UAL* |
| T2D 5[b,c] | F | 48 | 30.4 | 7.5 | 70 | *UAL* |
| T2D 6[b] | M | 61 | 27.4 | 7.1 | 90 | *Prodo* |
| T2D 7[c] | M | 59 | 27.7 | 6.5 | 85 | *Prodo* |
| T2D 8[c,e] | M | 63 | 22.0 | 7.3 | 80–85 | *Prodo* |
| T2D 9[c,e] | F | 43 | 35.8 | | 85 | *UAL* |
| T2D 10[c,e] | M | 42 | 43.7 | 6.6 | 95 | *Prodo* |
| T2D 11[c] | M | 53 | 30.1 | 7.8 | 80 | *Prodo* |
| T2D 12[c] | F | 37 | 33 | 13.1 | 85 | *Prodo* |
| **M = 9, F = 3** | | **52.67 ± 8.99** | **30.88 ± 6.25** | **8.0 ± 1.98** | | |

[a]Donors used for whole human islet studies synchronized by forskolin and collected around the clock.

[b]Donors used for whole human islet studies collected 12 h and 24 h after forskolin synchronization.

[c]Donors used for gene expression analysis by qPCR.

[d]Donors used for *siClock/siControl* experiments.

[e]Donors used for membrane fluidity assessment experiments.

Data in bold represent the mean values per group ± standard deviation.

BMI, body mass index; ND, nondiabetic; T2D, type 2 diabetes; M, male; F, female.

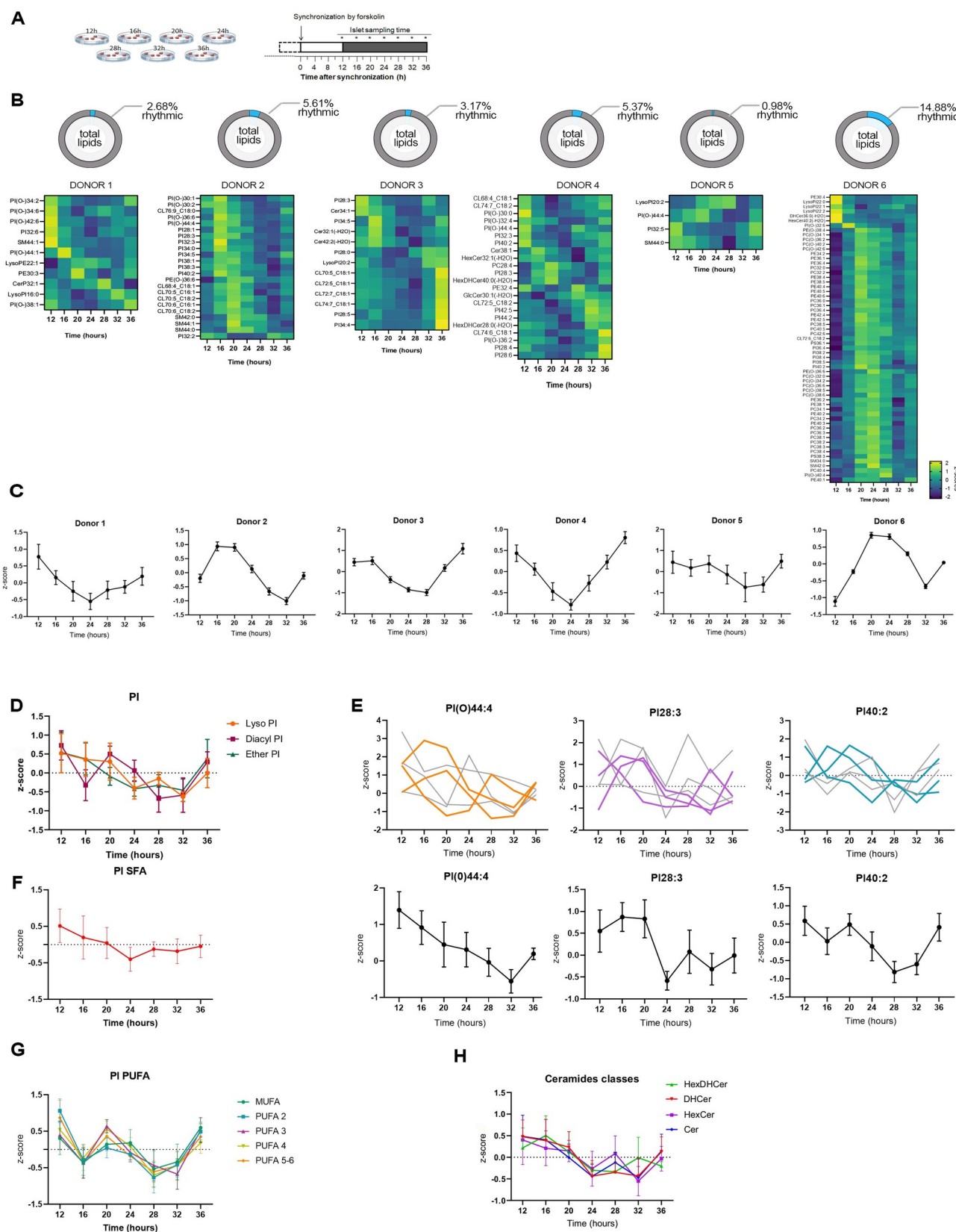

**Fig 1. Identification of rhythmic lipid metabolites in human islets.** (A) Experimental design for the collection of human pancreatic islets synchronized with forskolin pulse and harvested at the indicated 7 time points ($n = 6$ donors; asterisks indicate collection time). (B) Percentage of circadian rhythmic lipid species according to METACYCLE in pancreatic islets in each of the analyzed donors and corresponding heatmaps over the 7 time points. Normalized z-scores of lipid metabolites are indicated in yellow (high) and blue (low). See also S2 Data. (C) Average temporal levels of rhythmic lipids (normalized z-scores) shown in (B) for each islet donors. All profiles were qualified circadian rhythmic ($p < 0.05$), except for donor 5. Data are represented as mean ± SEM, $n$ = number of rhythmic lipids for each donor. See also S2 Data. (D) Average temporal profiles of PI species clustered by subclasses (Lyso-, Diacyl, Ether-PI). LysoPI abundance displays a significant circadian profile ($p < 0.05$). See also S1 Data. (E) Representative islet lipids identified as circadian rhythmic in 3 donors: PI(-O) 44:0, PI28:3, and PI40:2. Individual lipid profiles (top panels) with colored traces corresponding to the donors exhibiting a circadian rhythmic profile ($p$ value $< 0.05$); average lipid profiles (bottom panels). See also S1 Data. (F) Average temporal profiles of SFA PI phospholipid species. See also S1 Data. (G) Average temporal profiles of PI phospholipid species sorted by degree of saturation, from MUFA lipids to PUFA lipids. See also S1 Data. (H) Average temporal profiles of ceramide species by subclasses: HexDHCer, DHCer, HexCer, and Cer. DHCer abundance displays a significant circadian profile ($p < 0.05$). See also S1 Data. Lipid concentrations of PI shown in (F, G) were corrected for class II isotopic overlaps by performing additional deisotoping analysis on the normalized values. Data for (D–H) are represented as mean ± SEM, $n = 6$. See also S2 Fig. MUFA, monounsaturated fatty acid; PI, phosphatidylinositol; PUFA, polyunsaturated fatty acid; SFA, saturated fatty acid.

to what we observe for other lipid classes following synchronization (Figs 1F and 1G and S2I–S2K). Indeed, the phosphatidylcholine (PC), phosphatidylethanolamine (PE) and phosphatidylserine (PS), the most abundant classes of membrane lipids along with the PI, exhibited significant temporal changes of their levels across 24 h, but with no clear circadian pattern (S2I Fig). The PC/PE ratio, an indicator of cell membrane integrity, was relatively constant throughout the circadian cycle (S2L Fig).

In addition, we observed heterogenous temporal profiles of various sphingolipids (SL) (S2J Fig). Dihydroceramides exhibit a significantly rhythmic accumulation throughout the 24 h according to METACYCLE, and their temporal accumulation "around the clock" almost overlapped with the one of the ceramides (Cer), hexosylceramides (HexCer), and hexosyldihydroceramides (HexDHCer) (Fig 1H). This similar variation of abundance among the ceramides classes suggests a rhythmic de novo synthesis of the ceramides in pancreatic islets (Fig 1H). In contrast, the profile of the sphingomyelins (SM), the most abundant SL, was closer to that of the glycerophospholipids and cardiolipins (S2I–S2K Fig). Overall, lipid levels strongly changed over the course of 24 h, with variability observed among the islet donors. Importantly, the PI, and to a lesser extent, the Cer and HexCer exhibited oscillatory profiles throughout all the donors, suggesting a widespread impact of the circadian oscillator on these lipid classes metabolism in human pancreatic islets.

## Lipidomic profiling reveals major changes in lipid metabolites at 2 time points in human T2D pancreatic islets synchronized in vitro

After identifying circadian rhythmic lipid metabolites in the pancreatic islets from ND donors cultured and synchronized in vitro, we next attempted to measure their temporal alterations in T2D islets. Since we were not able to conduct a complete around the clock study on T2D human islets due to the lack of material, lipid profiles were assessed at 2 opposite time points, 12 h and 24 h following synchronization by forskolin pulse ($n = 5$ T2D donors) and compared to the ND islet counterpart ($n = 4$, see Table 1 and S3 Data). The selected time points correspond to peak and trough of the core clock gene *BMAL1* expression level (S1 Fig) and of the rhythmic profiles obtained for lipid species in most of the examined ND donors (4 out of 6 donors, Fig 1C). To assess whether T2D is characterized by global changes in lipid homeostasis in human islets, as it was the case in other metabolic tissues [35], we first averaged the levels of lipids detected at the 2 time points and compared those to the lipid class distribution in ND islets (Fig 2A–2D). Hierarchical clustering analysis of the top 30 lipid level changes shows an imperfect separation of the samples collected from T2D and ND donors, since the islet lipids from donor ND 10 clustered closer to the T2D counterpart than the other 3 ND individuals

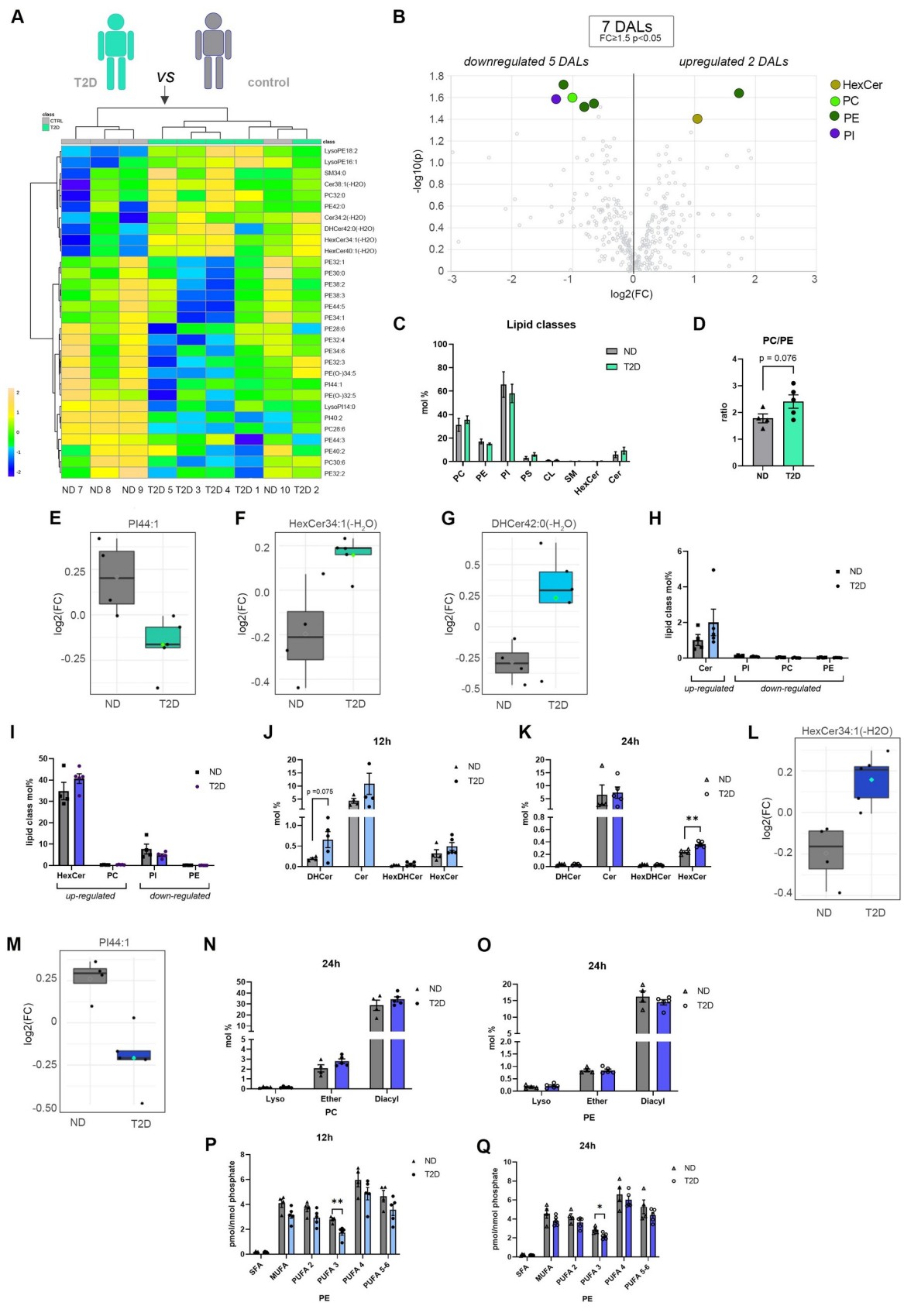

**Fig 2. Lipidomic analyses of human islets derived from T2D versus ND donors cultured and synchronized in vitro.** (A) Hierarchical clustering analysis (Distance Measure: Euclidian; Clustering algorithm: Ward) of top 30 islet lipids with most contrasting patterns between T2D and control ND counterpart. For each donor, islet lipids levels measured at 12 h and 24 h after forskolin synchronization were averaged ($n$ = 5 for the islets from T2D donors and $n$ = 4 for the islets from ND donors). (B) Volcano plots of differentially abundant islet lipids (fold change $\geq$ 1.5 and $p$ < 0.05, Welch's corrected) between T2D ($n$ = 5) and ND donors ($n$ = 4). Colored dots highlight significant up- or down-regulated individual lipid species. (C) Lipid class repartition (PC, PE, PI, PS, CL, HexCer, Cer, and SM) in human islets from ND and T2D donors (in mol%), synchronized in vitro and collected at 12 h and 24 h after in vitro synchronization. The data represent the average of the 2 time points ($n$ = 5 for the islets from T2D donors and $n$ = 4 for the islets from ND donors, mean ± SEM). (D) Comparison of the PC/PE ratio in islets from ND ($n$ = 4) versus islets from T2D donors ($n$ = 5). The data represent the average of the 2 time points (12 h and 24 h) ± SEM. (E, F) Representative examples of individual lipids (PI44:1 and HexCer34:1 (-H2O)) down- (E) and up- (F) regulated in the islets from T2D donors compared to islets from ND donors. The data represent the log2 fold change. (G) DHCer42:0(-H2O) levels in T2D versus ND islets synchronized in vitro and collected after 12 h. (H, I) Abundance of significantly differentially regulated lipids between the control and the T2D groups at 12 h (H) and 24 h (I). Each bar represents the sum of the significantly differentially regulated individual lipids shown in Fig 3E and 3F, as percentage of the total lipids detected from the same class at the same time point. Data are represented as mean ± SEM. (J, K) Relative level changes (mol%) of DHCer, Cer, HexDHCer, and HexCer in islets from T2D ($n$ = 5) and ND ($n$ = 4) donors collected 12 h (J) and 24 h (K) after synchronization, mean ± SEM. (L, M) Representative examples of individual lipids (HexCer34:1(-H2O) and PI44:1) up- (L) and down- (M) regulated in T2D versus ND islets synchronized in vitro and collected after 24 h. The data represent the log2 fold change. (N, O) Relative PC (N) and PE (O) level changes (mol%) in islets from T2D and ND donors collected 24 h after synchronization. Lipids are clustered according to the nature of the fatty acid linkage (diacyl versus alkyl-acyl (ether) or monoacyl (lyso)). T2D donors ($n$ = 5) and ND donors ($n$ = 4), mean ± SEM. (P, Q) Relative PE level changes (in pmol/nmol of phosphate with lipid concentrations corrected for class II isotopic overlaps) in islets from T2D and ND donors collected 12 h (P) and 24 h (Q) after synchronization represented according to the degree of saturation. T2D donors ($n$ = 5) and ND donors ($n$ = 4), mean ± SEM. Statistical analyses for (C, D, H–K, and N–Q) are unpaired $t$ test with Welch's correction. $^{*}p$ < 0.05, $^{**}p$ < 0.01. See also S3 Data. Cer, ceramide; CL, cardiolipin; DHCer, dihydroceramide; HexCer, hexosylceramide; HexDHCer, hexosyldihydroceramide; ND, nondiabetic; PC, phosphatidylcholine; PE, phosphatidylethanolamine; PI, phosphatidylinositol; PS, phosphatidylserine; SM, sphingomyelin; T2D, type 2 diabetes.

(Fig 2A). Overall, we observed a concomitant trend of decreased PE lipid levels and increased PC levels in the T2D islet group, resulting in a trend toward an increase PC/PE ratio that did not reach statistical significance (Fig 2C and 2D). Cer and HexCer exhibited a tendency toward increase in the T2D group, whereas several PI lipids were down-regulated (Fig 2A and 2B and 2E–2G). Because various phospholipid species (PE, PI, PC) were either up- or down-regulated in T2D islets (Fig 2A–2G), no significant change in overall amount per lipid class was observed between T2D and ND islets (Fig 2C).

Assuming that differences between the T2D and ND islets could be masked due to the average analysis across 2 time points, we next analyzed the islet lipids level at 12 h and 24 h separately. In this case, the hierarchical clustering showed a clear separation of the samples according to the donor group (T2D and ND) at both time points (Fig 3A and 3B), with a higher number of significant differentially abundant lipids (DALs) observed at 24 h (5 different lipids at 12 h versus 12 at 24 h, with fold change > 1.5 and $p$ < 0.05) (Fig 3C and 3D). None of the lipids differentially abundant between the groups was common across the 2 time points (Fig 3E and 3F), further highlighting the importance of temporal analysis, even if conducted in 2 time points only. Strikingly, several HexCer, representing approximately 40% of all lipids of this class (Fig 2H and 2I), were significantly differentially regulated between the T2D and ND groups (Fig 3F). Consistently, looking at the levels of all major lipid classes (Fig 3G and 3H), we noticed a higher level of total HexCer in the T2D group compared to the ND group at both time points with marked difference at 24 h (Figs 2J–2L and 3I). Several C16, C22, and C24 containing HexCer species were particularly increased 24 h after synchronization in the islets from the T2D donors compared to ND counterparts (Fig 3J). Whereas the ceramide levels were only slightly increased in T2D islets, abundant DHCer species were increased in the T2D groups at 12 h after synchronization (Fig 2H and 2J and 2K).

Among the phospholipids, few metabolites were down-regulated in the T2D group that mostly belonged to the PE and PI lipid classes (Figs 2M and 3E), in agreement with the previously observed global differences (Fig 2A and 2B). At 24 h, the concomitant increase of PC

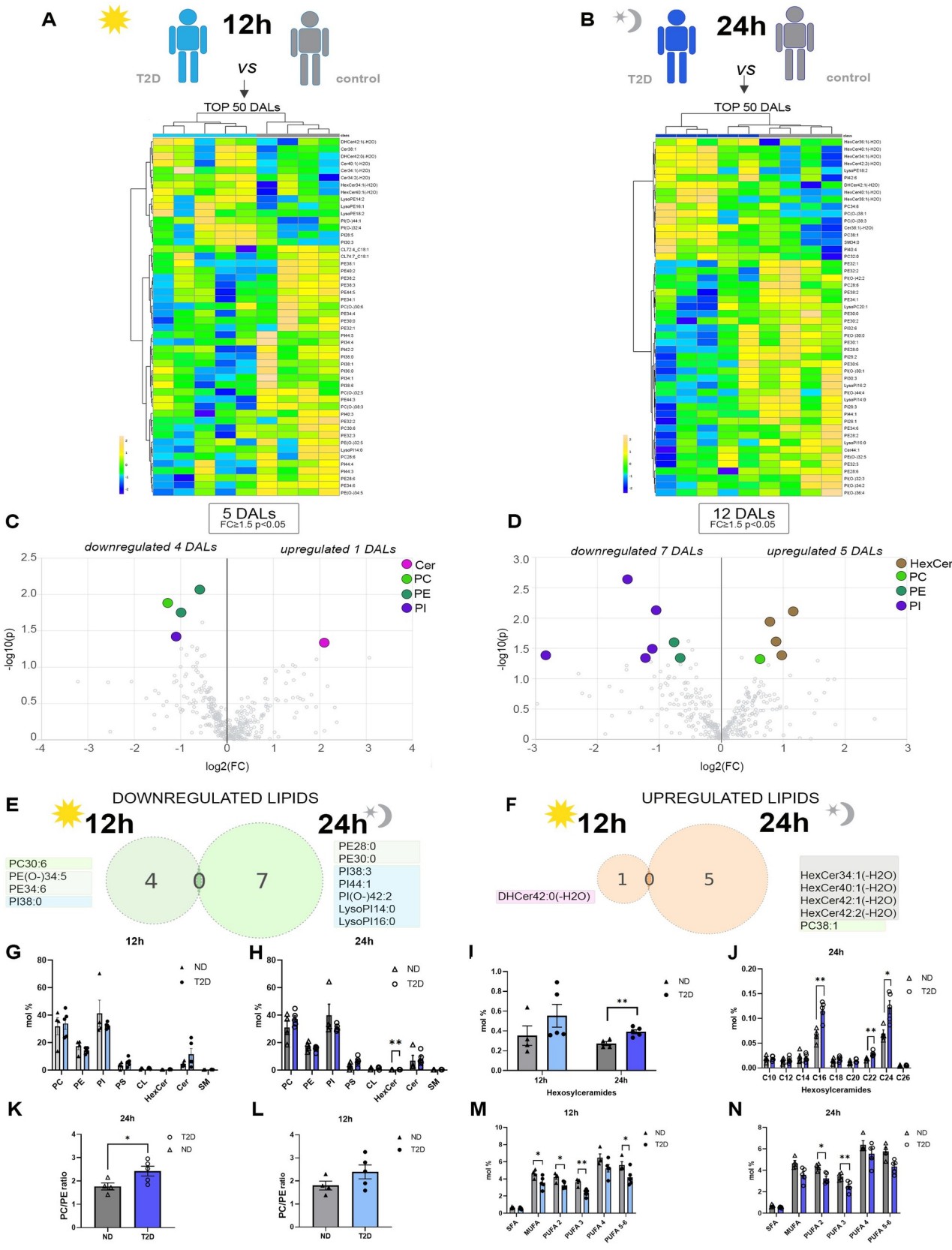

**Fig 3. Comparison of human islet lipidome from T2D and ND donors at 12 and 24 h after in vitro synchronization.** (A, B) Hierarchical clustering analysis (Distance Measure: Euclidian; Clustering algorithm: Ward) of top 50 islet lipids with most contrasting patterns between T2D and control patients at 12 h (A) and 24 h (B) after forskolin synchronization. (C, D) Volcano plots of differentially abundant islet lipids (fold change ≥ 1.5 and $p < 0.05$, Welch's corrected) at 12 h (C) and 24 h (D), between T2D and ND donors. Colored dots highlight significant up- or down-regulated individual lipid species. (E, F) Venn diagrams assessing the down-regulated (E) and up-regulated (F) lipid species shared by the 2 indicated time points in islets collected from T2D donors. (G, H) Lipid class repartition (PC, PE, PI, PS, CL, HexCer, Cer, and SM) in human islets from ND and T2D donors (in mol%), synchronized in vitro and collected at 12 h (G) and 24 h (H). (I) Comparison of the relative HexCer level changes (mol%) in ND versus T2D islets collected at 12 h and 24 h after synchronization. (J) Comparison of the relative HexCer level changes, sorted by number of carbons, over the total of islet lipids (mol%) measured in ND and T2D islets collected at 24 h after synchronization. (K, L) Ratio between PC and PE at 12 h (L) and 24 h (K) after synchronization. (M, N) Relative PE level changes (mol%) in islets from T2D and ND donors collected 12 h (M) and 24 h (N) after synchronization represented according to the degree of saturation. Statistics for (G–N) are unpaired 2-tailed $t$ test with Welch's correction. T2D donors ($n = 5$) and ND donors ($n = 4$), data are represented as mean ± SEM. $^*$ $p < 0.05$. $^{**}$ $p < 0.01$. See also S3 Data. Cer, ceramide; CL, cardiolipin; HexCer, hexosylceramide; ND, nondiabetic; PC, phosphatidylcholine; PE, phosphatidylethanolamine; PI, phosphatidylinositol; PS, phosphatidylserine; SM, sphingomyelin; T2D, type 2 diabetes.

and decrease of diacyl PE levels (Fig 2N and 2O) resulted in a significant increase of the PC/PE ratio, known to influence cellular calcium homeostasis and ER function [45] (Fig 3K). A similar change has been observed at 12 h; however, it did not reach statistical significance (Fig 3L). Whereas all unsaturated PE subspecies, regardless of their degree of saturation, exhibited the trend toward the decrease in the T2D group, this difference did not reach significance for PUFA 4 (at both time points) and PUFA 5 to 6 (at 24 h) (Fig 3M and 3N). In contrast, this difference was highly significant for PUFA 3, even after deisotoping correction of the lipid signals (Fig 2P and 2Q). Since increase in PUFAs within the membrane enhances membrane fluidity [46], the decrease in the PUFA-PE content might be indicative of defects in plasma membrane physical properties in the islets derived from T2D patients. Collectively, these experiments reveal major alterations in the pancreatic islet lipid homeostasis in T2D patients, potentially indicative of an increased inflammation and ER stress and reduced membrane plasticity.

## Diurnal ceramide levels correlate with transcript profiles encoding key enzymes involved in their turnover

Our data reveal that HexCer display both a rhythmic accumulation pattern around the clock in islets from ND patients synchronized in vitro and higher levels in islets from T2D patients. To explore the molecular determinants, we investigated the temporal gene expression profile of UDP-glucose ceramide glucosyltransferase (UGCG), a key enzyme involved in glucosylceramide biosynthesis that catalyzes the transfer of glucose from UDP-glucose to ceramide. Strikingly, the *UGCG* mRNA expression measured around the clock in the islets from ND donors exhibited a rhythmic profile ($p = 0.05$ as assessed by JTK_Cycle) with the trough around 20 to 24 h following in vitro synchronization, corroborating the temporal profile of *BMAL1* transcript (S1 Fig), and recapitulating the diurnal accumulation profile of HexCer (Fig 4A). Remarkably, we observe a concordance between the higher level of HexCer in the T2D group at 24 h and the *UGCG* transcript up-regulation in the same group compared to the ND group (Fig 4B).

We next assessed whether a correlation exists between the levels of Cer and DHCer lipid classes and the temporal mRNA profiles of ceramide synthases, involved in N-acylation of sphinganine and sphingosine bases to form DHCer and Cer. Ceramide synthase 2 (CerS2), the most abundant and ubiquitously expressed ceramide synthase [16,47–49] displays temporal variation in its mRNA expression that correlates with the ceramide and DHCer accumulation profiles (Fig 4C). In addition, the tendency for a higher amount of Cer and DHCer in islets from the T2D donors compared to their counterparts that was especially pronounced for DHCer at 12 h after synchronization, concordantly with the significant increase of *CerS2* transcript in T2D islets (Fig 4D).

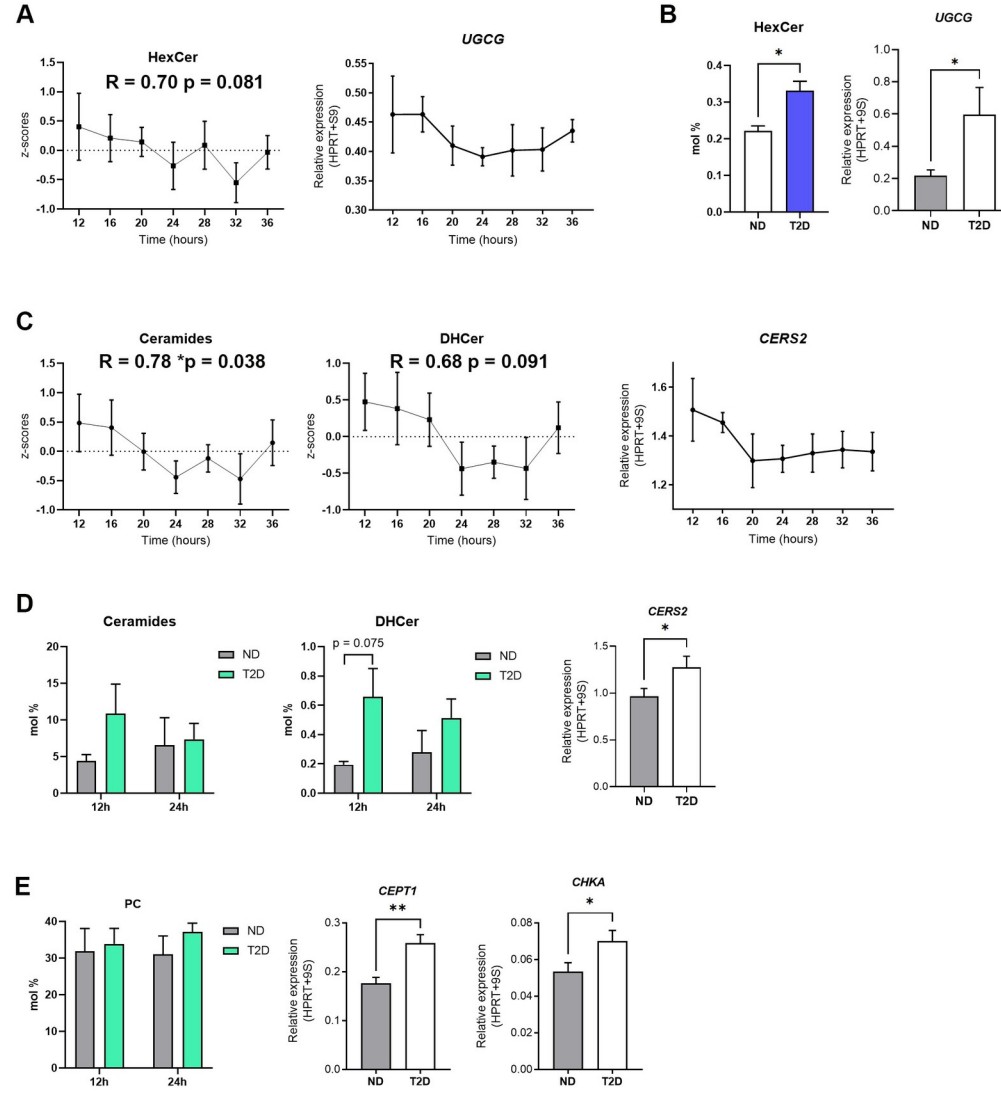

**Fig 4. Diurnal ceramide levels correlate with transcript profiles encoding for key enzymes involved in their biosynthesis.** (A) Average temporal profiles of HexCer species (left, $n = 6$) and RT-qPCR temporal gene expression profile of *UGCG* (right, $n = 3$). See also S1 and S4 Data. (B) Comparison of the relative HexCer level changes (mol%) in ND versus T2D islets collected at 24 h after in vitro synchronization (left, $n = 6$) with the *UGCG* expression measured in ND and T2D islets (right, $n = 9$). See also S3 and S4 Data. (C) Average temporal profiles of ceramide species (left, $n = 6$), DHCer (middle, $n = 6$), and RT-qPCR temporal gene expression profile of *CERS2* (right, $n = 3$). See also S1 and S4 Data. (D) Comparison of the relative ceramide (left) and DHCer (middle) level changes (mol%) in ND versus T2D islets collected 12 h and 24 h after synchronization ($n = 6$) with the *CERS2* expression measured in ND and T2D islets (right, $n = 9$). See also S3 and S4 Data. (E) Comparison of the relative levels of PC (mol%) in ND versus T2D islets collected 12 h and 24 h after synchronization (left, $n = 6$) with the *CEPT1* (middle) and *CHKA* (right) expression measured in ND and T2D islets ($n = 9$). See also S3 and S4 Data. The R numerical value indicates the corresponding Pearson correlation coefficient and its associated *p*-value. Transcript levels were normalized to *HPRT* and *9S* expression. Data are represented as mean ± SEM, unpaired 2-tailed *t* test; $p * < 0.05$, $p ** < 0.01$. DHCer, dihydroceramide; HexCer; hexosylceramide; HPRT, hypoxanthine-guanine phosphoribosyltransferase; ND, nondiabetic; PC, phosphatidylcholine; T2D, type 2 diabetes.

Beyond the enzymes involved in lipid metabolism that exhibited oscillatory patterns, we also analyzed the relationship between the enzymes that were significantly differentially expressed in islets from T2D compared to ND donors and the corresponding lipid class abundance in each group. *CEPT1* and *CHKA* genes code for choline/ethanolamine

phosphotransferase and choline kinase alpha enzymes, respectively, that are involved in PC biosynthesis via the CDP-choline pathway. Interestingly, the slight increase in PC levels in the T2D group, more pronounced at 24 h and contributing to a significant increase in the PC/PE ratio at this time point (Fig 3K), was associated with an up-regulation of both *CEPT1* and *CHKA* mRNA expression in the T2D group compared to the ND counterpart (Fig 4E).

## Lipid membrane fluidity of the human pancreatic islet cells is diminished in T2D islets and in ND islets upon circadian clock disruption

Collectively, the alterations in lipid metabolites that we observed in human pancreatic islets derived from T2D donors pointed toward a possibility of perturbed membrane organization and fluidity in T2D islet cells, which could impact their secretory function. We therefore measured membrane fluidity in human islet cells from T2D donors compared to ND counterpart (Fig 5A) by bioimaging using a Nile red derivative NR12S that is thought to penetrate only the outer leaflet of the plasma membrane [50]. Fluorescence emission of NR12S is sensitive to the membrane environment in a way that in more ordered membranes fluorescence emission is blue shifted, while in disordered membranes the fluorescence emission spectra is red shifted [50]. This shift in emission profile between liquid-disordered and liquid-ordered phases allows a quantitative assessment of membrane order by calculating the ratio of the fluorescence intensity recorded in 2 spectral channels, known as the generalized polarization (GP) value [51]. GP quantification of NR12S fluorescence emission from ND and T2D islet cell images revealed significant increase in membrane rigidity in T2D islet cells as compared to ND counterparts (Fig 5A).

We have recently demonstrated that functional perturbation of insulin secretion by human pancreatic islets derived from T2D islets was recapitulated in ND human islet cells upon cell-autonomous clock disruption, both in terms of diminished absolute secretion and perturbed rhythmic profile, indicating that islet cellular clock disruption may take part in pathophysiology of T2D in humans [13,15]. Whether such a parallel holds true for the changes in lipid homeostasis remains unexplored. Disruption of circadian oscillators in ND islet cells by transfection of siRNA targeting CLOCK following our previously validated protocols [13,52] led to significantly increased expression of *UGCG* (S3A Fig), similarly to the observed increase in this enzyme in T2D islets (Fig 4B). Moreover, KEGG pathway enrichment analysis of all significantly up-regulated transcripts in clock-compromised islets revealed activation of sphingolipid metabolism pathway ($p = 0.0575$; S3B Fig). To uncover changes in membrane fluidity in si*Clock*-transfected ND islet cells bearing disrupted oscillators, NR12S fluorescence has been compared between clock-compromised cells and control counterparts transfected with scrambled si*Control* RNA (Fig 5B, left). Strikingly, membrane fluidity was significantly reduced (Fig 5B, right), pinpointing that disruption of circadian oscillators in islet cells derived from ND donors leads to increased membrane rigidity, thus recapitulating the phenotype observed in T2D islets (compare Fig 5B to 5A).

## Perturbation of ceramide metabolism affects insulin secretion by human pancreatic islets and decreases lipid membrane fluidity

Given that major changes that we observed in lipid homeostasis in the islets from T2D donors were related to altered sphingolipid levels, we next assessed the impact of inhibiting ceramide de novo synthesis by myriocin on the islet function. Application of myriocin to ND islet cells resulted in greater GP values of NR12S emission as compared to nontreated control (Fig 5C), suggesting increase in membrane rigidity of these cells. To assess the effect of myriocin on induced insulin secretion by human pancreatic islets, we performed glucose-stimulated insulin

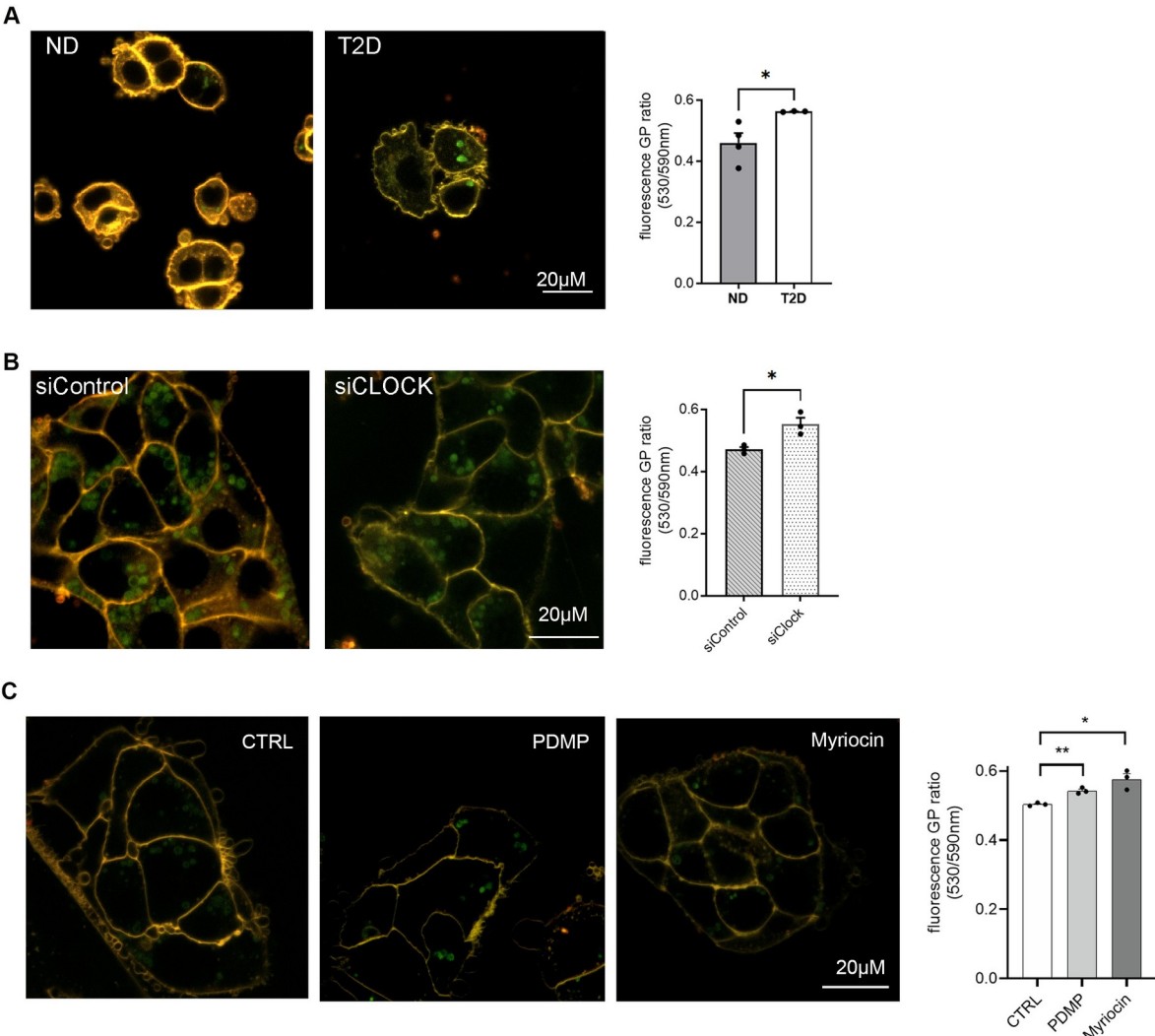

**Fig 5. Lipid membrane fluidity of the human pancreatic islet cells is attenuated in T2D islets, in ND islets upon siClock-mediated clock disruption, and in ND islets following perturbation of ceramide metabolism by PDMP or myriocin.** Live cell imaging of human islet cells stained with NR12S dye. (A) Representative ND (left) and T2D (right) human islet cells. The graph on the right summarizes the quantification of fluorescence GP ratio for $n$ = 4 ND and $n$ = 3 T2D donors. (B) Representative images of ND islet cells transfected with scrambled siRNA (*siControl*, left) bearing functional clocks and *siClock* targeting CLOCK protein that bear perturbed oscillators (right). The graph on the right summarizes the quantification of fluorescence GP ratio for $n$ = 3 donors. (C) Representative images of nontreated control ND islet cells (left), cells following 1-h treatment with PDMP (middle) or myriocin (right). The graph summarizes the quantification of fluorescence GP ratio for $n$ = 3 human donors. Note the significant up-regulation of membrane rigidity for T2D islet cells (A), ND cells with compromised clocks (B, *siClock*), and ND cells with impaired sphingolipid metabolism (C, PDMP and myriocin). Graph data are represented as mean ± SEM; unpaired 2-tailed $t$ test as compared to control, $p$ * < 0.05, $p$ ** < 0.01. See also S3 Fig and S4 Data. GP, generalized polarization; ND, nondiabetic; siRNA, small interfering RNA; T2D, type 2 diabetes.

secretion (GSIS) and KCL-stimulated insulin secretion (KSIS) tests. Application of myriocin to ND as well as to T2D islet cells led to compromised insulin secretion under low glucose conditions and a tendency to inhibiting both GSIS and KSIS that did not reach statistical significance (Fig 6A–6C). We next studied lipid membrane fluidity, GSIS and KSIS in the presence of PDMP that inhibits UCGC, the key enzyme of glycosphingolipid biosynthesis. Similar to myriocin, application of PDMP to ND islet cells significantly increased GP values of NR12S emission (Fig 5C). Strikingly, insulin secretion by ND islets was strongly compromised in the

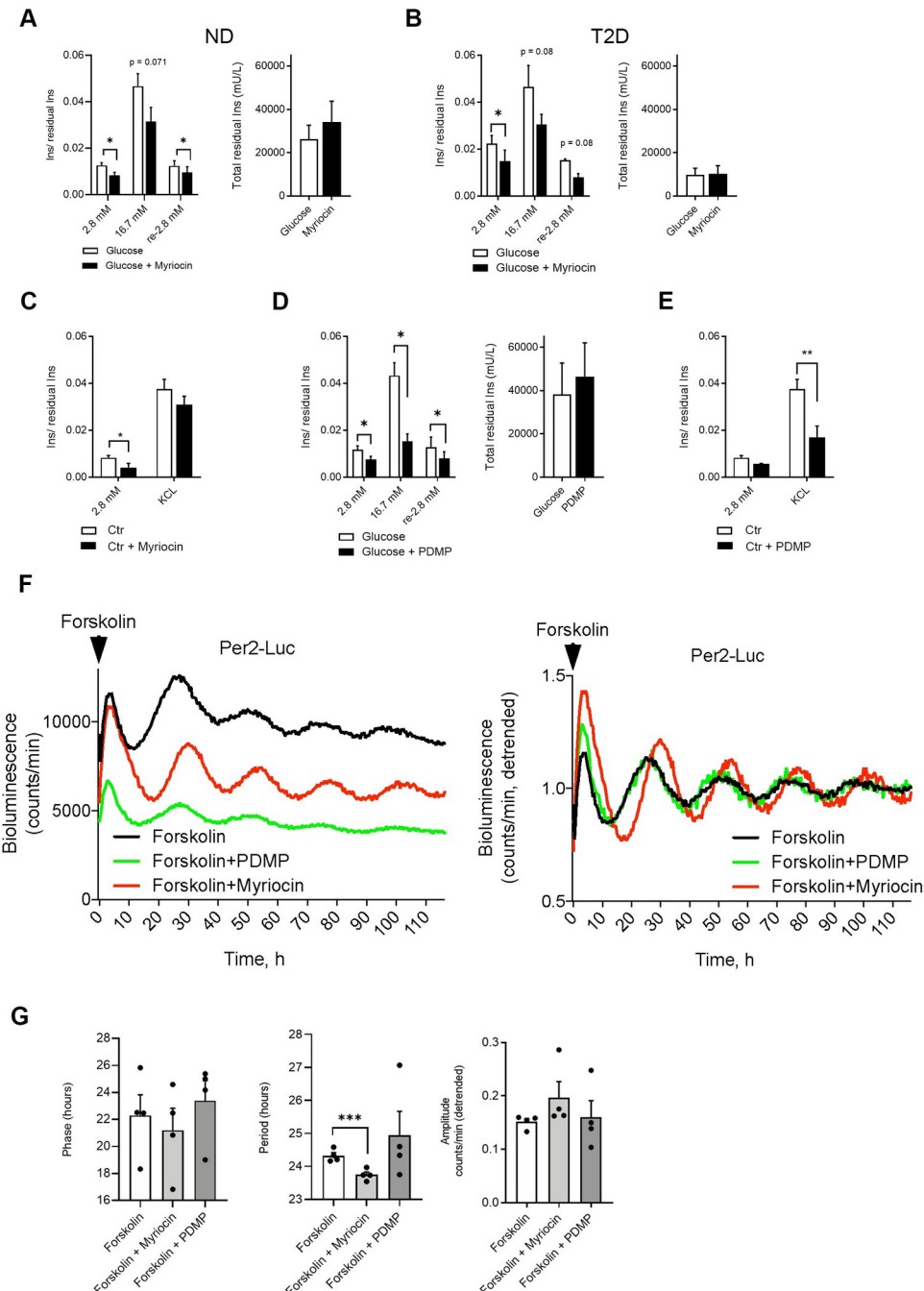

**Fig 6. Inhibitors of sphingolipid synthesis, PDMP and myriocin, perturb insulin secretion and circadian oscillations by human pancreatic islets.** Inhibitory effects of myriocin (A–C) and ceramide analog PDMP (D, E) on basal (1 h at 2.8 mmol glucose), glucose-induced (1 h at 16.7 mmol), and KCL-induced (1 h at 30 mmol KCL) insulin secretion in human islet cells in vitro. Data represent values normalized (Ins/residual Ins) to the total residual insulin content (total residual Ins (mU/L) presented on adjacent graphs) and are expressed as mean ± SEM for $n = 7$ ND donors (A), $n = 3$ T2D donors (B), $n = 3$ ND donors (C), $n = 5$ ND donors (D), and $n = 3$ ND donors (E). The difference is tested by paired 2-tailed $t$ test; $p^* < 0.05$, $p^{**} < 0.01$. See also S4 Fig. (F) Representative raw (left) and detrended (right) Per2-luc bioluminescence profiles of human islets in the presence of myriocin or PDMP during the entire bioluminescence recording. (G) Effect of myriocin and PDMP on principal circadian parameters (phase, period length, and amplitude) of Per2-luc oscillations. Data are represented as mean ± SEM, $n = 4$ ND donors. The difference is tested by 2-way ANOVA test with Bonferroni posttest; $p^* < 0.05$ (as compared to the control counterpart synchronized with forskolin in the absence of these compounds). See also S5 Fig and S4 Data. ND, nondiabetic; T2D, type 2 diabetes.

presence of PDMP at basal glucose levels, and after stimulation by high glucose, or by KCL (Fig 6D and 6E). A similar effect was observed when PDMP was applied to islet cells isolated from T2D patients (S4 Fig). Neither myriocin nor PDMP exerted a significant effect on the islet cell insulin content (Fig 6A and 6B and 6D, right graphs).

### Ceramide turnover inhibitors myriocin and PDMP alter circadian oscillations in human pancreatic islets synchronized in vitro

Since the ceramide levels exhibited circadian rhythmicity in ND islets on one hand and were strongly perturbed in T2D islets on the other, we next explored whether disrupted turnover of ceramides may feedback on the islet molecular clockwork. To this end, we recorded circadian bioluminescence of a Period2-luciferase (Per2-luc) lentiviral construct expressed in ND islet cells synchronized in vitro [14,15] in the presence of myriocin or PDMP in the recording medium (Fig 6F). Application of myriocin resulted in period shortening and phase advance of circadian oscillations of Per2-luc, while PDMP had no significant effect on the islet cell rhythmicity (Fig 6G). In contrast to myriocin that did not significantly affect cell mortality, PDMP showed a clear tendency to stimulate islet cell apoptosis following continuous 5-day exposure that did not reach statistical significance as compared to forskolin-treated control (S5 Fig).

## Discussion

Our study reveals that in human pancreatic islets derived from ND donors, about 5% of the lipid metabolites across all major lipid classes exhibited pronounced circadian oscillations following in vitro synchronization. In our previous lipidomic analysis, we report that in synchronized human primary myotubes, the circadian oscillating lipid species were more abundant, reaching up to 18.6% [29]. Such discrepancy may reflect tissue-specific lipid composition or stem from the high inter-donor variability, low number of the islet donors ($n$ = 6), limited amount of starting material, and islet cellular heterogeneity.

Among the different lipid species considered circadian in our analysis, we report major oscillations of PIs and SLs. PI metabolites are both components of cellular membranes and signaling molecules that are essential for secretory function of endocrine β-cells. Indeed, PI lipids generate soluble inositol second messengers involved in the mobilization of intracellular $Ca^{2+}$ stores and the recruitment of other signaling proteins regulating formation and secretion of granules at the plasma membrane [53]. Moreover, stimulation of insulin secretion influences PI metabolism in plasma membranes of MIN6 β-cell line [54,55]. We have previously shown that in vitro synchronized human islets exhibit a circadian profile of insulin secretion, with a respective peak and nadir appearing 12 h and 24 h after synchronization [13,15]. This circadian pattern of insulin secretion positively correlates with the temporal profile of PI abundance, also exhibiting a peak 12 h after forskolin pulse (Fig 1D). We speculate that the circadian oscillations of PI may participate in regulation of temporal secretion of insulin by β-cells. Moreover, islets derived from T2D donors exhibited a slight decrease in total PIs and a significant diminution of several individual abundant PI metabolites (Figs 2M and 3A–3E) concomitant with attenuated insulin secretion, further supporting a role of these PIs in the insulin release defects in diabetic β-cells.

Cer and HexCer are 2 additional major lipid classes exhibiting temporal variations in synchronized human islets, suggesting rhythmic organization of sphingolipid metabolism. Furthermore, the abundance of the total DHCer species exhibited circadian oscillations (Fig 1H). These temporal variations are correlated with the rhythmic transcription of *CERS2* and *UGCG*, the key enzymes involved in ceramide and glucosylceramide synthesis (Fig 4A and 4C), as well as with the expression of core clock transcripts (S1 Fig). Of note, in some instances,

mRNA measurements were conducted on the islets derived from different donors from those utilized for the lipidomics analyses (see Table 1). In line with these results, a large-scale RNA-seq screening of circadian transcripts in in vitro synchronized human islets reveal circadian rhythmic regulation of transcripts coding for key components of the sphingolipid metabolism (*CERKL*, *SGPL1*, *NEU2*, *NEU3*, *CERS6*, *CERS4*) [7], further highlighting a circadian regulation of this pathway at the transcriptional level. Concordantly, we also observed an up-regulation of *CERK*, *SGPL1* [13], and *UGCG* transcripts upon si*Clock* condition (S3 Fig), implying interconnection between the islet clockwork and regulation of ceramide synthesis. This finding is in line with our analysis of the human skeletal muscle lipid content upon CLOCK depletion [30] showing that *UGCG* expression is significantly up-regulated in si*Clock*-transfected primary myotubes compared to si*Control*-transfected ones. Accordingly, we observed a significant increase of the total HexCer in CLOCK-depleted skeletal muscle myotubes [29].

The interaction between molecular clock and sphingolipids seems to be bidirectional since decrease in sphingolipid levels by myriocin resulted in shortening of a circadian period length and phase advance in human islets (Fig 6F and 6G). Periodicity of clock machinery requires functional interactions of all its molecular components and depends on their expression and/or posttranscriptional modifications. For example, deletion or mutation of negative limb component *PER2* leads to a shortening of period length [56,57]. Nevertheless, our data indicate that myriocin failed to significantly down-regulate Per2-Luc expression, thus not supporting the idea that myriocin may exert its effect via modulation of PER2 absolute levels. On the other hand, phosphorylation of PER2 at different sites modulates the period length [58] and mutations associated with differential phosphorylation of human PER2 underlies familial advanced sleep phase syndrome [59]. Sphingolipids are considered to play an important role in intracellular signaling utilizing lipid–protein interactions [60,61]. Several protein candidates were shown to interact with ceramides, sphingosine 1 phosphate (S1P), and glycosphingolipids. Those comprise insulin receptor, ceramide-activated Ser-Thr phosphatases (PP1, PP2a), protein kinase B, protein kinase C zeta, and others [60–62]. It is not clear whether core clock components may be direct or indirect targets of sphingolipid species [58]. Further studies would be required to shed a light on the mechanisms underlying modulatory effect of sphingolipids on the molecular clock machinery.

Perturbation of islet sphingolipid metabolism takes part in pathogenesis of T2D early in disease development [53,63,64], as well as in T1D [65]. Due to the limited availability of human islets derived from T2D donors, we were unable to perform complete around the clock experiments for this part and compromised on 2-time point design. As a result, the temporal changes in lipid metabolites that are peaking at CT6 and CT18 were likely missed in T2D islets. The comparison of the lipid content between T2D and ND islets in 2 time points revealed an important modification of the sphingolipid fraction in the T2D islets (Fig 3). In line with altered levels of sphingolipids, expression of *CERS2* and *UGCG* was also up-regulated in T2D islets as compared to ND counterparts (Fig 4B and 4D). Ceramide accumulation, in particular in skeletal muscle and white adipose tissue, is associated with impaired insulin signaling and T2D [38,48,66]. However, in our case, the most striking difference between T2D and ND islet lipid content relies on the levels of HexCer, and to some extent on DHCer. UGCG enzyme located in Golgi is essential for the formation of glucosylceramides (GlcCer), precursors for most complex glycosphingolipids [67]. These glycosylated sphingolipids mainly localize in the external leaflet of the plasma membrane. They are involved in various cellular processes, including calcium homeostasis [68], membrane trafficking, and formation of membrane microdomains [69,70] that play important roles in the dynamic aggregation of membrane receptors, as demonstrated for the insulin receptor in mouse adipocytes [71,72]. Noteworthy, β-cell metabolic stress induced by acute palmitate treatment stimulates Cer production, while

longer (48 h) palmitate exposure increases, via the up-regulation of *UGCG*, the levels of GlcCer with no significant effect on SM and Cer accumulation, [42,73] thus recapitulating the changes we have detected in T2D human islets. Similarly, stress-induced Cer increase in keratinocytes following exposure to exogenous sphingomyelinase resulted in increased GlcCer synthase expression and GlcCer levels [74]. We hypothesize that enhancing the conversion of Cer into GlcCer, via the up-regulation of *UGCG*, prevents the deleterious effect of excessive Cer amounts and could thus reduce ER stress markers and apoptosis [74,75].

Here, we show that de novo production of sphingolipids is required for normal secretion of insulin by human islet cells in vitro, since application of sphingolipid synthesis inhibitor myriocin dampened the basal levels of secreted insulin in both ND and T2D human islet cells (Fig 6A–6C). Data in mouse islets and in rodent β-cell line further support an important role of sphingolipid metabolism for insulin secretion, since inhibition of this pathway in rodent β-cells by myriocin or fumonisin B1 attenuates insulin secretion [63,64]. At the same time, we showed that reduction of Cer by shunting them toward HexCer is necessary for proper basal and GSIS by human islet cells. Indeed, inhibition of GlcCer biosynthesis by PDMP significantly reduced basal insulin secretion and completely abolished their response to high glucose challenge or following KCL-triggered depolarization in vitro (Fig 6D and 6E). In addition to glucosylceramide synthase inhibition, the attenuation of mTOR signaling pathway and lysosomal lipid accumulation reported following PDMP treatment [76] may partly account for its inhibitory effect on insulin secretion. Importantly, knockdown of *UGCG* by siRNA in mouse islets also resulted in major insulin secretion defects [77], further suggesting that UGCG plays a key role in the observed phenotype.

Recent studies showed that pharmacological inhibition of 2 other ceramide-shunting pathways (sphingomyelin biosynthesis by D609 and S1P biosynthesis by sphingosine-kinase inhibitor SKI), similarly to described here HexCer biosynthesis inhibition by PDMP, significantly reduces basal and glucose-induced insulin secretion in vitro as well as in vivo in rodents [63,64]. In MIN6 cells, glucose stimulation enhanced conversion of Cer to GlcCer and to SM [78], as well as accumulation of S1P but not Cer [79]. Together, these data suggest that proper Cer homeostasis is required for stimulus-secretion coupling in β-cells.

Noteworthy, several DHCer species were increased in T2D islets compared to ND islets, with a marked difference 12 h after synchronization (Figs 2A, 2G, 2J, 3A, 3C, and 3F). The significant increase of DHCer42:0(-$H_2O$) (C24DHCer) observed in the T2D islets may mask an up-regulation of C24DoxCer. We and others recently reported that noncanonical 1-deoxyceramide (DoxCer) levels were elevated in serum and adipose tissue of T2D patients [35,80]. Given that DoxCer has the exact same *m/z* ratio as DHCer-$H_2O$, these 2 lipid species may be misidentified, thus requiring a separate assessment using liquid chromatography mass spectrometry to be properly measured [35]. Deoxysphingolipids were shown to compromise GSIS in rodent islets and Ins-1 cells [34] allowing to envisage a specific role of these toxic sphingolipids in the failure of pancreatic β-cells. Further lipidomic analyses in human islets should be conducted to conclude this link in humans.

Importantly, we demonstrate that the plasma membrane of T2D pancreatic islet cells is more rigid compared to ND counterparts (Fig 5A). Lowered membrane fluidity was reported in erythrocytes [81], leukocytes [82], and platelets [83,84] from T2D patients, suggesting a potential generality of membrane stiffness upon T2D. Strikingly, a similar phenotype was observed in clock-compromised islet cells from ND donors that exhibited stiffer plasma membrane than their counterparts bearing functional clocks (Fig 5B) and following direct sphingolipid perturbation by PDMP or myriocin (Fig 5C). Rheological properties of the membrane bilayer rely on lipid composition and cholesterol content [85,86]. Thus, saturated lipids increase membrane rigidity, whereas polyunsaturated phospholipids that bear more flexible

chains facilitate membrane conformational state changes by increasing membrane flexibility and fission [85,87–90]. Consequently, membrane PUFAs might be particularly critical for cells that go through multiple endocytic events such as epithelial cells [89]. Our lipidomic analysis revealed an overall trend for lowering PE species, with the levels of PE-PUFAs being significantly decreased in islets from T2D patients as compared to their ND counterparts (Figs 2P and 2Q and 3M and 3N). At the same time, the levels of PC lipids stayed relatively stable, thus resulting in increased PC/PE ratio that reaches significance at 24 h after synchronization (Fig 3K). Membrane fluidity impacts on cell communication with the environment by affecting the receptor function, signal transduction, endo-, and exocytosis. Indeed, decreased membrane fluidity reduces the insulin signaling in kidney mononuclear leukocytes and in diabetic kidney [82,91]. The exact mechanism of membrane fluidity changes in the clock-compromised and T2D human islet cells and its role on insulin secretion and signal transduction in β-cells needs to be assessed in future studies.

In summary, our large-scale lipidomic analyses provide the first systematic characterization of the temporal organization of lipid metabolite landscape in human pancreatic islets from ND donors. Our recent study demonstrated that molecular clocks are compromised in pancreatic islets from T2D human donors, leading to disrupted absolute and temporal profiles of insulin and glucagon secretion [15]. Here, we report time-of-the-day-specific alterations of lipid metabolism in T2D human islets. The changes in lipid composition and saturation degree were concomitant with observed decrease of membrane fluidity in T2D human islets. Strikingly, a drop-in membrane fluidity was recapitulated in ND islets bearing compromised clocks, in line with a similar parallel between disrupted islet hormone secretion between T2D and ND islets transfected with *siClock* in our previous study [15]. Finally, our data suggest a reciprocal connection between the islet circadian clocks and Cer metabolism. Perturbation of Cer turnover observed in human pancreatic islets upon T2D may lead to exacerbation of the islet clock disruption and thus to further disturbance of lipid homeostasis in a feed-forward loop. Altogether, we provide a novel link between disruption of circadian clock, temporal coordination of lipid metabolism in human pancreatic islet, and islet dysfunction upon T2D in humans, highlighting both molecular oscillator and sphingolipid metabolites as important regulators of insulin secretion and membrane fluidity.

## Material and methods

### Pancreatic islet and islet cell culture

Human pancreatic islets were obtained from 4 different sources, summarized in Table 1: (i) Prodo Laboratories company (ND and T2D islets); (ii) Alberta Diabetes Institute islet core center (UAL) (ND and T2D islets); and (iii) Islet Transplantation Center of Geneva University Hospital (ND islets). T2D donors had a history of T2D and/or HbA$_{1c}$ greater than 6.5%. Details of the islet donors are summarized in Table 1. All procedures using human islets were approved by the ethical committee of Geneva University Hospital CCER 2017–00147. Human pancreatic islets were cultured in CMRL 1066 medium, containing 5.5 mM glucose and supplemented with 10% fetal bovine serum (Gibco), 110 U/ml penicillin (Gibco), 110 µg/ml streptomycin (Gibco), 50 µg/ml gentamicin (Gibco), 2 mM L-glutamine (GlutaMax, Gibco), and 1 mM sodium pyruvate (Gibco). Islet cell gentle dissociation was done using 0.05% Trypsin (Gibco) treatment. For lipidomic analysis, approximately 600 islets were plated to 35-mm dishes (Falcon). For bioluminescence recordings, 100 islets were plated to multi-well plates (LifeSystemDesign). For the rest of the experiments, approximately 50,000 dissociated islet cells were attached to 35-mm dishes (Falcon). All dishes were precoated with a homemade laminin-5-rich extracellular matrix derived from 804G cells as described in [92].

## Viral transduction and small interfering RNA transfection

Human islet cells were transduced with Per2-luc lentivectors as described in [14]. Dissociated adherent human islet cells were transfected twice with 50 nM siClock or with the same amount of nontargeting siControl (Dharmacon, GE Healthcare, Little Chalfont, United Kingdom) [13,52].

## In vitro cell synchronization and circadian bioluminescence recording

Adherent islets were synchronized by a 1-h pulse of forskolin (10 μM; Sigma, Saint-Louis, Missouri, United States of America) with a subsequent medium change. The islets were subjected to continuous bioluminescence recording in CMRL medium containing 100 μM luciferin (D-luciferin 306–250, NanoLight Technology) during at least 5 consecutive days. For the experiments with ceramide and hexosylceramide biosynthesis inhibitors, 100 nM N-[2-hydroxy-1-(4-morpholinylmethyl)-2-phenylethyl]-decanamide, monohydrochloride (PDMP, Cayman Chemical) or 100 nM myriocin (Sigma-Aldrich), respectively, were applied together with forskolin synchronization pulse and added to the recording medium for the entire experiment duration. Bioluminescence pattern was monitored by a homemade robotic device equipped with photomultiplier tube detector assemblies, allowing the recording of 24-well plates [93] or by LumiCycle 96 (Actimetrics). In order to analyze the amplitude, period length, and phase of time series without the variability of magnitudes, raw data were processed in parallel graphs by moving average with a window of 24 h [13]. Cell apoptosis was assessed where indicated with Cell Death Detection ELISA kit (Roche) in the end of bioluminescence recording experiments, according to manufacturer instructions.

## Lipid extraction procedures

The lipidomic extractions were performed as described in [35]. A total of 600 human islets were harvested (approximately $6 \times 10^5$ cells) at the indicated time points after 1-h pulse of forskolin synchronization (Fig 1A) or as indicated otherwise (Figs 2 and 3) and resuspended in 100 μL $H_2O$. Lipid extracts were prepared using a modified MTBE (methyl-tert-butyl ether) extraction protocol with addition of internal lipid standards [94]. Briefly, 360 μL methanol and a mix of internal standards were added (400 pmol PC 12:0/12:0, 1,000 pmol PE 17:0/14:1, 1,000 pmol PI 17:0/14:1, 3,300 pmol PS 17:0/14:1, 2,500 pmol SM d18:1/12:0, 500 pmol Cer d18:1/17:0, and 100 pmol GlcCer d18:1/8:0). After addition of 1.2 mL of MTBE, samples were placed for 10 min on a multitube vortexer at 4°C followed by incubation for 1 h at room temperature (RT) on a shaker. Phase separation was induced by addition of 200 μL MS-grade water. After 10 min at RT, samples were centrifuged at 1,000g for 10 min. The upper (organic) phase was transferred into a 13-mm glass tube, and the lower phase was re-extracted with 400 μL artificial upper phase [MTBE/methanol/$H_2O$ (10:3:1.5, v/v/v)]. The combined organic phases were separated into 2 aliquots and dried in a vacuum concentrator (CentriVap, Labconco). Phospholipids were eluted with methanol (3 × 500 μL) and divided into 2 aliquots. One aliquot was used for glycerophospholipid and phosphorus assay, respectively, while the other aliquot was treated by mild alkaline hydrolysis to enrich for sphingolipids, according to the method by Clarke [95]. Briefly, 1 mL freshly prepared monomethylamine reagent [methylamine/$H_2O$/n-butanol/methanol (5:3:1:4, (v/v/v/v)] was added to the dried lipid extract and then incubated at 53°C for 1 h in a water bath. Lipids were cooled to RT and then dried. For desalting, the dried lipid extract was resuspended in 300 μL water-saturated n-butanol and then extracted with 150 μL $H_2O$. The organic phase was collected, and the aqueous phase was re-extracted twice with 300 μL water-saturated n-butanol. The organic phases were pooled and dried in a vacuum concentrator.

## Determination of total phosphorus

Total phosphorus was determined as described in [35]. Briefly, 100 μL of the total lipid extract, resuspended in chloroform/methanol (1:1), were placed into 13-mm disposable pyrex tubes and dried in a vacuum concentrator, and 0, 2, 5, 10, 20 μL of a 3 mmol/L $KH_2PO_4$ standard solution were placed into separate tubes. To each tube, distilled water was added to reach 20 μL of aqueous solution. After addition of 140 μL 70% perchloric acid, samples were heated at 180°C for 1 h in a chemical hood. Then, 800 μL of a freshly prepared solution of water, ammonium molybdate (100 mg/8 mL $H_2O$), and ascorbic acid (100 mg/6 mL $H_2O$) in a ratio of 5:2:1 (v/v/v) were added. Tubes were heated at 100°C for 5 min and cooled at RT for 5 min. Approximately 100 μL of each sample was then transferred into a 96-well microplate, and the absorbance at 820 nm was measured.

## Phospho- and sphingolipid analysis by mass spectrometry

Mass spectrometry analysis was performed using multiple reaction monitoring on a TSQ Vantage Extended Mass Range Mass Spectrometer (Thermo Fisher Scientific), equipped with a robotic nanoflow ion source (Triversa Nanomate, Advion Biosciences) as previously described [35]. Optimized fragmentation was generated using appropriate collision energies and s-lens values for each lipid class. Mass spectrometry data were acquired with TSQ Tune 2.6 SP1 and treated with Xcalibur 4.0 QF2 software (Thermo Fisher Scientific). Lipid quantification was carried out using an analysis platform for lipidomics data hosted at EPFL Lausanne Switzerland (http://lipidomes.epfl.ch/). Quantification procedure was described in [96]. Dried lipid extracts were resuspended in 250 μL MS-grade chloroform/methanol (1:1) and further diluted in either chloroform/methanol (1:2) plus 5 mmol/L ammonium acetate (negative ion mode) or in chloroform/methanol/H2O (2:7:1) plus 5 mmol/L ammonium acetate (positive ion mode).

## RNA extraction and qPCR

Total RNA was prepared from cultured islet cells using RNeasy Plus Micro Kit (Qiagen). The RNA concentration was measured by Qubit RNA SH kit (Invitrogen). Then, 0.2 μg of total RNA was reverse-transcribed using Superscript II (Invitrogen) and random hexamers and was PCR-amplified on a LightCycler 480 (Roche Diagnostics AG, Basel, Switzerland). Mean values for each sample were calculated from technical duplicates of each quantitative RT-PCR (qRT-PCR) analysis and normalized to the mean of housekeeping genes hypoxanthine-guanine phosphoribosyltransferase (*HPRT*) and *S9*. Primers used for this study are listed in S1 Table.

## Membrane fluidity experiments

For assessment of membrane fluidity, the islets cells were seeded onto a glass-bottom dishes (Willco) at a density of 30,000 cells/dish. For microscopy imaging, the attached cells were washed once with warm CMRL 1066 medium with no phenol red (Gibco-Invitrogen), supplemented with 2 mM L-glutamine (GlutaMax, Gibco), 1 mM sodium pyruvate (Gibco), and 15 mM HEPES to maintain pH. After that, 200 μL of a 2 μM NS12R dye solution (Klymchenko Laboratory [50]) diluted in the same medium was added, and the cells were incubated for 5 min at RT. Cells were washed 3 times with the warm medium and immediately subjected to fluorescence microscopy using a Nikon A1r microscope, equipped with CFI Plan Apo ×60 oil (NA = 1.4) objective. The excitation in confocal mode was provided by a 488-nm laser, while the fluorescence was detected at 2 spectral ranges: 550 to 600 ($I_{550-600}$) and 600 to 650 ($I_{600-650}$)

nm in sequential mode by rapid switching to minimize drift. All the parameters at each channel were left constant. The laser power was set at 1% of maximum intensity to achieve a good signal. At least 10 confocal images were recorded using NIS Elements per 1 dish. The fluorescence shift radiometry was assessed using Fiji after outlaying membrane area. GP was calculated as follows: $GP = (I_{550-600} - gI_{600-650})/(I_{550-600} + gI_{600-650})$. Where coefficient g was previously calculated for the NR12S solution in CMRL.

## Data quantification and analyses

Lipid concentrations were calculated relative to the relevant internal standards and normalized to the total phosphate content of each total lipid extract (S1–S3 Data). Then, for comparison between different lipids samples, relative lipid concentrations were normalized to the total lipid content of each lipid extract (mol%). For temporal analysis, normalized lipid values were z-scored within patients. To identify circadian variations within the lipidomic data set, normalized lipid values were further analyzed using the METACYCLE v1.2.0 algorithm in R Bioconductor v3.11 [97]. The period width was set to fit a time frame of 20 to 28 h and a $p$ value of $\leq 0.05$ was considered statistically significant.

Lipid concentrations were corrected for class II isotopic overlaps for the analysis of lipid degree of saturation as described in [98]. Briefly, correction factors for deisotoping were derived using theoretical M+2 abundances calculated using the Envipat Web 2.4 tool (https://www.envipat.eawag.ch/) applying a mass resolution of 5,000. These theoretical M+2 abundances were multiplied by a correction factor accounting for the probability at random distribution of two $^{13}C$ isotopes within the remaining heavy fragment generated during the fragmentation in the collision chamber (Q2) but not detected in the Q3. The resulting formula for correction is: $M + 2_{correction} = (M + 2_{theoretical}) \times ((n_{heavy})/(m_{total}))^2$ with $n_{heavy}$ being the number of carbons in the heavy fragment, and $m_{total}$, the number of carbons in the entire lipid molecule. For each lipid species, the corrected $M + 2$ signal was calculated and subtracted from the acquired signal for the lipid species with $m/z + 2$ within a series of lipid species from the same lipid class, beginning with the most desaturated species, stepwise until reaching the fully saturated form.

Additional data processing (filtering, normalization, transformation, scaling), statistical analyses, and data plotting were performed using MetaboAnalyst 5.0 [97] and Prism Graph Pad 8.0. Statistical tests used for comparison between groups are indicated in the figure legends. Differences were considered significant for $p \leq 0.05$ (*), $p \leq 0.01$ (**), and $p \leq 0.001$ (***).

To determine the clustering, k-NN (nearest neighbors with k clusters) was applied to the phases and amplitudes in polar coordinates of all circadian signals for k = 1, 2, and 3 clusters.

## Supporting information

**S1 Fig. Oscillations of core clock and clock-related genes in forskolin-synchronized human islets.**
(DOCX)

**S2 Fig. Lipidomic analyses of human islets derived from T2D versus ND donors cultured and synchronized in vitro.**
(DOCX)

**S3 Fig. Up-regulation of sphingolipid metabolism in clock-deficient human islet cells.**
(DOCX)

**S4 Fig. PDMP inhibits basal (1 h at 2.8 mmol glucose) and glucose-induced (1 h at 16.7 mmol) insulin secretion in human islet cells from T2D donor in vitro ($n = 1$).**
(DOCX)

**S5 Fig. Assessment of apoptosis in pancreatic islets treated with myriocin or PDMP during continuous bioluminescence recordings for 5 days.**
(DOCX)

**S1 Table. Sequences of quantitative RT-PCR primers.**
(DOCX)

**S1 Data. Lipidomic data regarding the human pancreatic islets from ND donors synchronized in vitro.**
(XLSX)

**S2 Data. Rhythmic islet lipids according to METACYCLE algorithm ($p < 0.05$).**
(XLSX)

**S3 Data. Lipidomic data regarding the human pancreatic islets from ND and T2D donors collected 12 h and 24 h after synchronization.**
(XLSX)

**S4 Data. Raw data for Figs 4–6.** Naming of sheets corresponds to the relevant figures. Data for assigned figure panels on sheet is introduced with corresponding legend.
(XLSX)

**S5 Data. Raw data for S1 and S3–S5 Figs.** Naming of sheets corresponds to the relevant supplemental figures. Data for assigned figure panels on sheet is introduced with corresponding legend.
(XLSX)

## Acknowledgments

The authors thank Domenico Bosco and Thierry Berney (Islet Transplantation Center of Geneva University Hospital), Eduard Montanya Mias and Montserrat Nacher Garcia (Hospital Universitari de Bellvitge, Barcelona) and Alberta Diabetes Institute IsletCore for providing human islets; Giovanni d'Angelo (EPFL) for support with the lipidomic analyses and for constructive discussions; François Prodon and Bioimaging platform team, Faculty of Medicine, University of Geneva; Andrey Klymchenko (CNRS, Strasbourg) for generously sharing Nile Red dye NR12S.

## Author Contributions

**Conceptualization:** Volodymyr Petrenko, Flore Sinturel, Howard Riezman, Charna Dibner.

**Data curation:** Volodymyr Petrenko, Flore Sinturel, Ursula Loizides-Mangold, Jonathan Paz Montoya, Simona Chera, Charna Dibner.

**Formal analysis:** Volodymyr Petrenko, Flore Sinturel, Ursula Loizides-Mangold, Jonathan Paz Montoya, Simona Chera.

**Funding acquisition:** Volodymyr Petrenko, Howard Riezman, Charna Dibner.

**Investigation:** Volodymyr Petrenko, Flore Sinturel, Ursula Loizides-Mangold, Jonathan Paz Montoya, Simona Chera, Charna Dibner.

**Methodology:** Volodymyr Petrenko, Flore Sinturel, Ursula Loizides-Mangold, Jonathan Paz Montoya, Howard Riezman.

**Project administration:** Howard Riezman, Charna Dibner.

**Resources:** Charna Dibner.

**Software:** Simona Chera.

**Supervision:** Howard Riezman, Charna Dibner.

**Validation:** Volodymyr Petrenko, Flore Sinturel, Ursula Loizides-Mangold, Jonathan Paz Montoya, Simona Chera, Howard Riezman, Charna Dibner.

**Visualization:** Volodymyr Petrenko, Flore Sinturel, Simona Chera.

**Writing – original draft:** Volodymyr Petrenko, Flore Sinturel, Charna Dibner.

**Writing – review & editing:** Volodymyr Petrenko, Flore Sinturel, Ursula Loizides-Mangold, Jonathan Paz Montoya, Simona Chera, Howard Riezman, Charna Dibner.

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
