## [Editor Report · Decision Letter 0]

3 Feb 2022

Dear Dr Dibner, 

Thank you for submitting your manuscript entitled "Circadian orchestration of lipid metabolism and membrane fluidity in human pancreatic islets are disrupted upon type 2 diabetes" for consideration as a Research Article by PLOS Biology.

Your manuscript has now been evaluated by the PLOS Biology editorial staff as well as by an academic editor with relevant expertise and I am writing to let you know that we would like to send your submission out for external peer review.

Once your full submission is complete, your paper will undergo a series of checks in preparation for peer review. Once your manuscript has passed the checks it will be sent out for review. To provide the metadata for your submission, please Login to Editorial Manager (https://www.editorialmanager.com/pbiology) within two working days, i.e. by Feb 07 2022 11:59PM.

If your manuscript has been previously reviewed at another journal, PLOS Biology is willing to work with those reviews in order to avoid re-starting the process. Submission of the previous reviews is entirely optional and our ability to use them effectively will depend on the willingness of the previous journal to confirm the content of the reports and share the reviewer identities. Please note that we reserve the right to invite additional reviewers if we consider that additional/independent reviewers are needed, although we aim to avoid this as far as possible. In our experience, working with previous reviews does save time. 

If you would like to send previous reviewer reports to us, please email me at ialvarez-garcia@plos.org to let me know, including the name of the previous journal and the manuscript ID the study was given, as well as attaching a point-by-point response to reviewers that details how you have or plan to address the reviewers' concerns. 

Given the disruptions resulting from the ongoing COVID-19 pandemic, please expect some delays in the editorial process. We apologise in advance for any inconvenience caused and will do our best to minimize impact as far as possible.

Kind regards,

Ines

--

Ines Alvarez-Garcia, PhD

Senior Editor

PLOS Biology

---

## [Decision Letter · Decision Letter 1]

28 Mar 2022

Dear Dr Dibner,

Thank you for submitting your manuscript entitled "Circadian orchestration of lipid metabolism and membrane fluidity in human pancreatic islets are disrupted upon type 2 diabetes" for consideration as a Research Article at PLOS Biology. Thank you also for your patience as we completed our editorial process, and please accept my apologies for the delay in providing you with our decision. Your manuscript has been evaluated by the PLOS Biology editors, an Academic Editor with relevant expertise, and by three independent reviewers.

You will see that the reviewers find the conclusions novel and interesting, however they each raise several points that should be clarified or amended to strengthen the results of the manuscript. Reviewers 1 and 2 ask mainly for clarifications, whereas Reviewer 3 raises two points that require the generation of additional data, but we think they can be easily addressed.

In light of the reviews (attached below), we are pleased to offer you the opportunity to address the comments from the reviewers in a revised version that we anticipate should not take you very long. We will then assess your revised manuscript and your response to the reviewers' comments and we may consult the reviewers again.

We expect to receive your revised manuscript within 1 month.

**IMPORTANT - SUBMITTING YOUR REVISION**

3. Resubmission Checklist

a) *PLOS Data Policy*

b) *Published Peer Review*

d) *Blurb*

Please also provide a blurb which (if accepted) will be included in our weekly and monthly Electronic Table of Contents, sent out to readers of PLOS Biology, and may be used to promote your article in social media. The blurb should be about 30-40 words long and is subject to editorial changes. It should, without exaggeration, entice people to read your manuscript. It should not be redundant with the title and should not contain acronyms or abbreviations. For examples, view our author guidelines: https://journals.plos.org/plosbiology/s/revising-your-manuscript#loc-blurb

Sincerely,

Ines

--

Ines Alvarez-Garcia, PhD

Senior Editor

PLOS Biology

Reviewers' comments

Rev. 1:

In this manuscript, Petrenko et al examine the lipidome of pancreatic islet cells from normal donors (ND) and type-II diabetes (T2D) subjects. They demonstrate a number of lipid species and classes are oscillatory ex-vivo in isolated cells from NDs, as well as alterations in glycerophospholipids (GPLs) and sphingolipids in T2Ds between two timepoints. A strength of the study is the further investigation of these lipid alterations by examination of key enzymes of sphingolipid regulation which showed rhythmicity in NDs, and disruption by clock perturbation in NDs as well as in T2Ds. T2Ds also demonstrated reduced membrane fluidity similar to clock-disrupted NDs. Finally the authors demonstrate reduced glucose induced insulin secretion from inhibition of glycosphingolipid synthesis, while inhibiting earlier step ceramide synthesis elicits a weaker response. The authors concluded that that the circadian clock is required for temporal control of lipid homeostasis and disruption of the clock impairs insulin secretion.

Overall the work is clearly written and easy to follow, and the experimental approach is logical. Unfortunately the quality of the figures in the pdf I had was quite low and the figures were difficult to evaluate.

A few limitations or clarifications which should be discussed:

1. The lipidomics experiments were performed on cells from six NDs, however these individuals span a dramatic range in BMI (23.7-44.1). Given the fairly small number of NDs in this experiment, this has an important impacts on the overall lipid cycling and the ability to detect cycling lipids. As the authors note from the Chua et al, individual lipid profiles in blood demonstrate immense variability in cycling. While the approach to select for lipids cycling in 3/6 NDs mitigates this to some extent, the combination of low numbers and heterogenous population is likely hampering discovery of potentially important cycling lipids and / or classes. Was other clinical chemistry information available from these donors such as TGs or other lipid panels? In summary, the total number of cycling lipids is probably vastly underestimated.

2. Some explanation for why the ND cells used for lipidomics were not the same as those used for the qPCR of enzyme analysis is required. Given the concerns above, this is particularly important as there is no guarantee that the inter-individual variability does not extend to these enzymes (and hence a potential mismatch cycling between the lipids and enzymes). In general, the use of different donors for the different aspects of the study is a weakness (although I understand that acquiring sufficient material may be an issue).

3. Why were 12 and 24 h chosen as the time points for the T2D lipidomics analysis? There are a range of peaks phases apparent in Supp Fig 1C, and while a number peak at 12, these appear to be different lipid classes for each donor (assuming the colors relate to the classes in Supp Fig 1B; its not made clear). I think some aggregate analysis of phase properties would be appropriate to include in Figure 1. In addition, some discussion of those lipids / lipid classes that peak between ZT 6 or 18 is worth mentioning as these are likely not going to be capture in the current design.

4. Lines 255-258: The authors suggest that the increase in UGCG mRNA is indicative of glucosylceramide production as opposed to galactosylceramides due to a simarity in profile. Other circadian studies have suggested that the lag between mRNA expression and protein expression averages to ~6 h (Mauvoisin et al, 2014; Robles et al, 2014) and that in certain pathways the lag to metabolites is ~12 h (Krishnaiah et al, 2017). While there is wide variance in this literature data, the temporal nature of protein expression and function does not support the claim being made in the current manuscript in my opinion. A similar association is suggested on lines 262-269 for Ceramide synthase 2.

5. The authors use PDMP to inhibit UCGC and demonstrate a significant impact on glucose induced insulin secretion. PDMP has been shown to have other effects on cells in addition to impacting glycolipid synthesis, including alterations on membrane structure and cholesterol homeostasis. The potential glycophosolipid-independent effects of PDMP on GSIS should be discussed.

Other comments:

6. The numbers of donors for each experiment should be specified in the abstract for clarity.

Rev. 2:

In this manuscript by Petrenko et al, the authors explore the circadian control of lipid species and membrane fluidity in ND and T2D islets. The authors demonstrate that lipid species profiles vary on a daily basis and that these profiles are altered in T2D islets. They further demonstrate that membrane fluidity is altered in both T2D islets and in islets where CLOCK has been knocked down. Additionally, it is demonstrated that disruption of de novo ceramide synthesis and glycosphingolipid biosynthesis impair insulin secretion and that ceramide turnover feeds back on circadian oscillations in the islets. Overall, the authors conclude that a functional circadian clock is required for the temporal control of lipid homeostasis and that disruption of lipid metabolism in T2D contributes to altered membrane fluidity and ultimately impaired GSIS.

This is a well-written and clearly presented manuscript. The data is convincing, the approaches are clearly described and the conclusions drawn are supported by the data presented. This reviewer only has minor comments/suggestions to provide a few points of clarity.

1. In table 1, the term "Gender" is used in the second column. It appears the authors meant to refer to biological sex as genders are not actually provided in this table. That said, this column should be changed to "sex" and notably, Male (M) and Female (F) are the typical terms used, not M and W

2. Line 178 - this subtitle wording is awkward. Suggest something like: "Lipidomic profiling reveals major changes in lipid metabolites at two time points in synchronized human T2D islets"

3. Line 181: Should be "to measure" not measuring

4. While this reviewer recognizes that journals have limits to the number of figures, there are many figures in the supplemental figures that should be considered for use in the primary figures. For example, on page 8 in the results section, the entire first paragraph on the lipidomic profiling section refers entirely to supplemental data. I would suggest that the authors reconsider the data in the supplemental figures and when there are significant differences that are discussed in the results, these figures should be presented in the primary figures.

5. Results line 206-208: "Consistently, we noticed a higher level of total hexosylceramides in the T2D group compared to the ND group at both time points…". This is not obvious in the graph presented in 2H. The authors should consider including an inset that only has this condition on a smaller scale so that the difference can be better visualized.

6. Figure 2K, L: It would be helpful to include PC/PE ratio on the x-axis rather than just stating ratio.

7. Results line 218-219: "…all unsaturated PE subspecies were decreased in the T2D group, regardless of their degree of saturation (Fig 2M-N)". This statement doesn't seem fully supported by the data in figure 2N as PUFA4 and 5-6 don't seem to show a significant reduction in T2D, as presented in the figure. Consider re-wording this statement.

8. Subtitle on line 249-250: This subtitle is a little awkward. Suggest removing the word "for" from the subtitle.

9. For qPCR data in Figure 3 - how was the data normalized? It is understood from the materials and methods that the data is expressed relative to a housekeeping gene, but was this data at 24 hours expressed relative to the data at 12 hours in each condition?

10. Discussion line 463: "…myriocin dampened the levels of secreted insulin in both ND and T2D human islet cells". This statement seems to be a bit of an over statement as myriocin did not significantly impair GSIS at 16.7mM only insulin secretion at 2.8mM was significantly affected.

11. Discussion Lines 466-480: How do you reconcile the increased apoptosis in the PDMP treatment condition? This should also be discussed and considered in the interpretation of the "impaired" GSIS results.

Rev. 3:

In this manuscript Petrenko and colleagues follow up on a previous body of work detailing transcriptional and functional properties of human islets that are subject to circadian clock control in vitro by attempting to mechanistically probe the function of rhythmic phospholipid and sphingolipid metabolism in glucose-stimulated insulin secretion. They apply lipidomic analysis to non-diabetic, type 2 diabetic and siCLOCK treated human islets and show that specific subsets of phospholipids and ceramide metabolite precursors of sphingolipids accumulate rhythmically and are altered by CLOCK silencing and diabetes. They link clock transcription-factor mediated control of lipids perturbed in diabetic islets to islet function by demonstrating that perturbing glucosylceramide synthesis (by increasing UGCG transcript) inhibits insulin secretion, possibly by altering the disposal of toxic ceramide species or influencing membrane fluidity. Importantly, this manuscript elaborates a layer of circadian control occurring at the level of lipid homeostasis in a key glucose-regulatory tissue that has been understudied, particularly in primary human cells. I think the work presented here is novel and will benefit and may facilitate further intellectual exchange between the circadian, diabetes and metabolic physiology research communities. However, I have several suggestions to strengthen the manuscript:

Major points:

1. The authors demonstrate that ceramides and genes involved in ceramide and sphingolipid metabolism are rhythmic and increase in the setting of CLOCK inhibition or diabetes, however there are a number of possibly addressable missing links between clock transcription factors and lipid metabolism-related enzymes. First, it would help if the authors would superimpose or compare the RNA expression profiles of transcriptional activators (CLOCK, BMAL1, DBP) and repressors (PERs, CRYs, REVERB, E4BP4) with key lipid-biosynthetic enzymes pointed out throughout the manuscript. This would help in understanding how the circadian clock network is regulating lipid composition. Next, the authors should compare the expression of CLOCK and other components of the clock transcription loop in their T2D and siCLOCK samples to determine the extent to which changes in expression of clock transcription regulators might underlie lipid changes in the T2D samples. Lastly, given that many of the ceramide synthesis genes were up-regulated in T2D and UGCG transcript is increased in siCLOCK islets it would be useful to determine if any known circadian repressors are up-regulated in these settings and/or if canonical circadian repressor motifs are present in the promoters of these genes.

2. The hypothesis that lipid mis-regulation in T2D and by clock disruption inhibits insulin secretion by increasing membrane rigidity is compelling, however a missing piece of data that would strengthen this argument would be to show that PDMP or myriocin treatment increases the fluorescence GP ratio. However, if the contribution of ceramides and sphingolipids to GSIS occurs independently of membrane dynamics the authors should explain this more thoroughly. It is also surprising that myriocin (ceramide turnover inhibitor) treatment reduces basal insulin secretion while PDMP (UGCG activator) reduces glucose-stimulated insulin secretion. It could be useful to compare effects of myriocin and PDMP on non-metabolic secretagogue-stimulated insulin secretion (such as KCl) to exclude impacts of metabolic and second messenger signaling pathways contributing to the mobilization and exocytosis of insulin vesicles.

Minor points:

1. The authors note that a number of phospholipid classes displayed non-circadian temporal profiles (p.5), raising the possibility that their synthesis might be related to cAMP induction caused by the forskolin pulse used to synchronize the islets. Do these temporal profiles display an induction early in the time course followed by a steady decline after washout of forskolin?

2. T2D islets were assayed at 12 and 24 hours after forskolin pulse (p. 8) but a rationale for these time points was not chosen. Please state why these times were selected.

3. In figure S4 PDMP further down-regulates GSIS in islets from a T2D donor. The authors should explain why this might be occurring since a hypothesis is that increased UGCG activity underlies circadian and diabetes-induced secretory impairment. Is the extent of suppression similar or different to ND islets and does PDMP suppress GSIS in siCLOCK islets?

4. In figure 6 panel A it appears the data are mislabeled. The colors for myriocin and PDMP are switched for the left and right panels.

5. Since human islets contain a relatively higher proportion of glucagon secreting alpha cells and the methods employed here do not appear to discriminate between alpha and beta cells the authors should comment on the possibility that circadian-induced changes in lipid metabolism and membrane dynamics occur in alpha cells impacting glucagon secretion. Since they have previously generated circadian RNA-seq data (in mouse) in alpha and beta cells it would be useful to know if key lipid metabolism genes are differentially controlled by the clock in these cell populations.

6. In figure 4A only two cells can be seen in the field for each condition. If possible the authors should choose a more convincing image.

7. Figure S3B shows pathways with p values typically higher than what is traditionally used in gene ontology. The case for these pathways being overrepresented could be strengthened by listing the number of genes represented in each of these pathways overlapping with DEGs in siCLOCK islets.

---

## [Editor Report · Decision Letter 2]

27 May 2022

Dear Dr Dibner,

Thank you for your patience while we considered your revision entitled "Circadian orchestration of lipid metabolism and membrane fluidity in human pancreatic islets are disrupted upon type 2 diabetes" for publication as a Research Article at PLOS Biology. This revised version of your manuscript has been evaluated by the PLOS Biology editors and the Academic Editor.

Based on this assessment, we are likely to accept this manuscript for publication, provided you satisfactorily address the following data and other policy-related requests.

In addition, we would like you to consider a suggestion to improve the title:

"Type 2 diabetes disrupts circadian orchestration of lipid metabolism and membrane fluidity in human pancreatic islets"

We expect to receive your revised manuscript within two weeks. 

*Published Peer Review History*

*Press*

Sincerely,

Ines

--

Ines Alvarez-Garcia, PhD

Senior Editor

PLOS Biology

Fig. 1B-H; Fig. 2A-D, G-N; Fig. 3A-Q; Fig. 4A-E; Fig. 5A-C; Fig. 6A-G; Fig. S1; Fig. S2B, C, E-L; Fig. S3A, B; Fig. S4 and Fig. S5

BLURB

Please also provide a blurb which (if accepted) will be included in our weekly and monthly Electronic Table of Contents, sent out to readers of PLOS Biology, and may be used to promote your article in social media. The blurb should be about 30-40 words long and is subject to editorial changes. It should, without exaggeration, entice people to read your manuscript. It should not be redundant with the title and should not contain acronyms or abbreviations. For examples, view our author guidelines: https://journals.plos.org/plosbiology/s/revising-your-manuscript#loc-blurb

---

## [Editor Report · Decision Letter 3]

24 Jun 2022

Dear Dr Dibner,

Thank you for the submission of your revised Research Article entitled "Type 2 diabetes disrupts circadian orchestration of lipid metabolism and membrane fluidity in human pancreatic islets" for publication in PLOS Biology. 

On behalf of my colleagues and the Academic Editor, Samer Hattar, I am very happy to say that we can in principle accept your manuscript for publication, provided you address any remaining formatting and reporting issues. These will be detailed in an email you should receive within 2-3 business days from our colleagues in the journal operations team; no action is required from you until then. Please note that we will not be able to formally accept your manuscript and schedule it for publication until you have completed any requested changes.

PRESS

Sincerely, 

Ines

--

Ines Alvarez-Garcia, PhD

Senior Editor

PLOS Biology
